# Drug and single-cell gene expression integration identifies sensitive and resistant glioblastoma cell populations

Robert K. Suter [1] ✉, Anna M. Jermakowicz[1], Rithvik Veeramachaneni[1], Matthew D'Antuono [1], Longwei Zhang [1], Rishika Chowdary[1], Simon Kaeppeli [1], Madison Sharp[1], Pravallika Palwai[1], Vasileios Stathias[2,3], Grace Baker[1], Luz Ruiz[1], Winston Walters[4], Maria Cepero[5], Danielle Burgenske[6], Edward B. Reilly[7], Anatol Oleksijew[7], Mark G. Anderson[7], Sion Ll. Williams[3,5], Michael E. Ivan [3,4], Ricardo J. Komotar[3,4], Macarena I. De La Fuente[3,5], Gregory Stein [8], Alexandre Wojcinski[9], Santosh Kesari[9,10], Jann N. Sarkaria [6], Stephan C. Schürer [2,3] & Nagi G. Ayad[1] ✉

Glioblastoma (GBM) remains the most common and lethal adult malignant primary brain cancer with few treatment options. A significant issue hindering GBM therapeutic development is intratumor heterogeneity and plasticity. GBM tumors contain neoplastic cells within a fluid spectrum of diverse transcriptional states. Identifying effective therapeutics requires a platform that predicts the differential sensitivity and resistance of these states to various treatments. Here, we develop scFOCAL (**S**ingle-**C**ell **F**ramework for -**O**mics **C**onnectivity and **A**nalysis via **L**1000), to quantify the cellular drug sensitivity and resistance landscape. Using single-cell RNA sequencing of newly diagnosed and recurrent GBM tumors, we identify compounds from the LINCS L1000 database with transcriptional response signatures selectively discordant with distinct GBM cell states, and leverage this capability to predict combination synergy. We validate the significance of these findings in vitro, ex vivo, and in vivo, and identify a combination of an OLIG2 inhibitor and Depatux-M for the treatment of GBM. Our studies suggest that scFOCAL identifies cell states that are sensitive and resistant to targeted therapies in GBM using a measure of cell and drug connectivity, which can be applied to identify new synergistic combinations.

Glioblastoma (GBM) remains the most common malignant adult brain cancer, with a median overall survival of only 15 months[1–3]. No new targeted therapies have been approved for GBM since the introduction of the alkylating agent temozolomide (TMZ) in 2005[1–3]. Thus, novel therapeutic options are direly needed for patients with GBM.

Intratumor heterogeneity poses a significant barrier to identifying effective targeted therapies for GBM[4,5]. Single GBM tumors contain diverse populations of cells with considerable genomic, transcriptomic, and proteomic differences[6–9]. GBM single-cell characterization has revealed a plastic transcriptional spectrum reminiscent of canonical neurodevelopmental cell types[6]. These transcriptional states encompass astrocyte-like (AC-like), neural-progenitor-like (NPC-like), oligodendrocyte-progenitor-like (OPC-like), and mesenchymal-like (MES-like) cells. The importance of their relative abundance is

highlighted by the identification of transcriptional subtype (Classical, Proneural, Mesenchymal) plasticity and switching following standard-of-care treatment[10]. Therefore, it is necessary to develop novel means of targeting these dynamic tumor cell populations. However, there is no current method to predict which compounds target the different neoplastic cell states within GBM tumors. We and others have shown that cancer gene expression-based disease signature reversal is predictive of compound efficacy[11–13]. Using the NIH Library for Integrated Network-based Cellular Signatures (LINCS) L1000 assay dataset, we previously developed transcriptional consensus signatures (TCSs) for each small molecule, consisting of genes that are consistently up- or down-regulated irrespective of cell type. We showed that effective combination treatments can be identified through the integration of compound TCSs with TCGA-derived GBM disease signatures and that compound efficacy in a GBM subtype-specific manner can be predicted[12–14]. However, these analyses do not address the GBM intratumor heterogeneity that is evident after performing single-cell RNA sequencing (scRNAseq) to predict mechanisms of drug sensitivity or resistance.

Here, we describe a framework for the integration of scRNAseq data and L1000 TCSs to facilitate in silico perturbation analyses informing combination therapy design, scFOCAL (**S**ingle-**C**ell **F**ramework for -**O**mics **C**onnectivity and **A**nalysis via **L**1000). Using scFOCAL, we demonstrate that the integration of GBM single-cell disease signatures with LINCS L1000-derived compound response signatures can identify drugs most likely to target transcriptionally distinct cell populations in individual GBM tumors. Furthermore, we demonstrate that this framework predicts the sensitive and resistant cell populations within GBM tumors in an orthotopic xenograft model. Using patient single-cell RNA sequencing data and a TCS for the aurora kinase inhibitor alisertib, we predict and confirm both the MES-like transcriptional identity of a distinct alisertib-resistant cell population and the depletion of transcriptionally NPC-like cells in vivo[15]. Moreover, we use publicly available datasets to demonstrate the utility of this approach to predict treatment-induced transcriptional response in discrete GBM cell populations and to identify synergistic small molecule combinations. Building on this, we leverage our framework to identify a combination of the OLIG2 inhibitor, CT-179, with an anti-EGFR antibody-drug conjugate, Depatuxizumab Mafodotin (Depatux-M, ABT-414), which synergizes to increase survival in an orthotopic xenograft model of GBM. Importantly, we introduce scFOCAL as a publicly available R package and Shiny application[16]. scFOCAL uses the methods defined here to generate and analyze cell-type-specific disease signatures, perform in silico drug connectivity analyses, characterize cell population sensitivity or resistance to specific treatments, and prioritize synergistic combinations.

## Results

To identify the different cell types within GBM tumors, we performed single-cell RNA-sequencing of six tumors resected from three newly diagnosed and three recurrent GBM patients. We combine our sequencing results with those from published studies that use the same 10X Genomics platform to study GBM at the single cell level, thereby increasing sample size for our downstream analyses (Fig. 1)[17]. Our dataset (hereto referred to as Suter et al.) and the Johnson et al. dataset are integrated and assessed using CytoTRACE, revealing a large tumor cell population marked by transcriptional diversity and de-differentiation (Fig. 1d)[18]. This population contained tumor cells spanning the neurodevelopmentally-rooted GBM cell transcriptional states identified by Neftel et al. (Fig. 1e, f)[6]. In addition to neoplastic cells, both datasets contain populations of myeloid cells, T-cells, fibroblasts, endothelial cells, and oligodendrocytes. These non-neoplastic cell populations that comprise the tumor microenvironment (TME) are identified by the expression of specific marker genes, such as *PTPRC* for myeloid cells and *CD3E* for T-cells (Supplementary

Fig. 1), and contain cells captured across both datasets (Fig. 1g). We then designate the transcriptional profiles of these TME cells as our normal cell reference to be able to compare the distinct GBM tumor cell states represented within our dataset. Utilizing this comparison, we were able to derive a pseudo-bulk disease signature for each of the cell states within GBM tumors (Supplementary Fig. 1h). The disease signatures unique to each transcriptional state reflected previously reported expression markers, such as *CDK4* and *SOX4* overexpression in NPC- and OPC-like cells, *EGFR* overexpression in AC-like cells, and *CD44* overexpression in MES-like cells (Supplementary Fig. 1h).

## Integration of L1000 TCSs with expression data of single GBM cells clusters, small molecules by mechanism of action

While distinct GBM single-cell transcriptional states have unique disease signatures relative to TME cell controls, this method relies on prior stratification and knowledge of tumor cell populations. We therefore seek a way to integrate L1000 TCSs with the expression of individual cells. We use Spearman's ρ between a single cell's normalized and scaled expression and TCS signatures to score compounds for their connectivity with each cell in the dataset, as done previously using bulk gene expression data (Fig. 2a)[19,20]. Through correlation analysis of each compound's calculated connectivity across the single-cell landscape, we found that many compounds cluster by their mechanism of action based on their tumor-cell connectivities (Fig. 2b, c, and Supplementary Fig. 2a, b). Compound classes such as MEK, PI3K, and mTOR, HSP, HDAC, and BET inhibitors cluster individually based on their correlations with GBM tumor single-cell expression data (Pearson's ρ > 0.7), suggesting that biologically relevant information is retained in this perspective (Fig. 2c). Interestingly, when looking only at 64 FDA-approved oncology drugs identified using annotations from the Drug Repurposing Hub[21], distinct clusters of small molecules can be identified based on their calculated connectivity with all cells within the GBM scRNAseq atlas (Supplementary Fig. 2a, b).

## The integration of L1000 TCSs of FDA-approved oncology drugs with the expression data of single GBM cells identifies distinct drug connectivity states within single tumors

To investigate whether drug connectivity patterns correlate with previously published GBM cell transcriptional states, we perform a clustering analysis of single cells from individual tumors based on their connectivity to 64 different FDA-approved drugs (Fig. 2d–g). Within individual tumors, clear patterns of distinct drug connectivity states are revealed, with some individuals containing at least 7 different states, where the number of unique states was determined using the ideal number of clusters k based on within-cluster sum of squares (Supplementary Fig. 3a). These patient-level drug connectivity clusters show patterns in their makeup by Neftel states, developmental potential quantified using CytoTRACE, and cell cycle phase (Fig. 2f, g, column annotations, and Supplementary Fig. 3a)[18]. Hierarchical clustering of all resulting patient clusters on their associated Neftel et al. state proportions, cell cycle proportions, and mean CytoTRACE score reveals an organization into distinct, high-level meta clusters—namely a proliferative cluster, an NPC-like cluster, an NPC/OPC-like cluster, an AC-like cluster, and a MES-like cluster (Supplementary Fig. 3b).

## Integration of L1000 TCSs with single-cell-derived disease signatures predicts differential sensitivity to small molecules

We seek to determine whether the identified pseudo-bulk disease signatures for AC-like, MES-like, OPC-like, and NPC-like states within GBM tumors can be targeted with distinct small molecules or drugs. To accomplish this, we use a previously reported method of disease signature reversal scoring[13]. In brief, a disease-signature-specific discordance score was calculated for all compounds within the L1000 library as the number of genes perturbed by the compound TCS in the

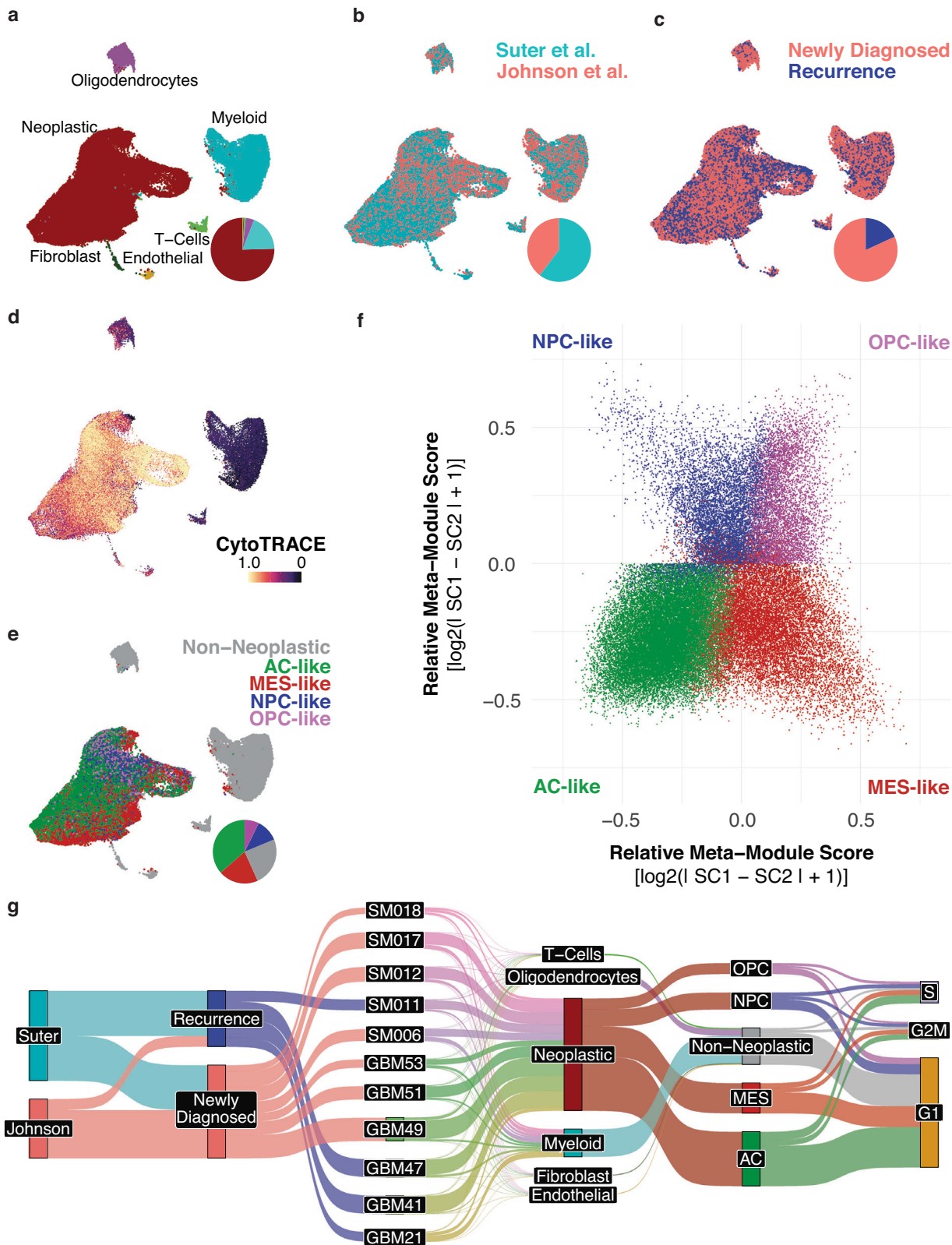

**Fig. 1 | Single-cell RNA sequencing reveals distinct transcriptional states present in newly diagnosed and recurrent GBM.** Single-cell RNA sequencing data of 6 patient glioblastoma tumors was integrated and harmonized with an external dataset obtained from Johnson et al. (2021)[17] **a**–**e**. UMAPs of single-cell transcriptomes colored by **a** cell type, **b** source dataset, **c** newly diagnosed or recurrent tumor status, **d** CytoTRACE score, and **e** Neftel et al. (2019) GBM cell transcriptional state assignment. **f** Two-dimensional representation of relative enrichment of GBM cell transcriptional states in neoplastic cells. Cells are colored by assigned transcriptional state identity based on the most predominantly enriched signature of that cell. **g** Sankey plot depicting proportions of cells grouped by source dataset, occurrence or recurrence, tumor ID, cell type, GBM cell transcriptional state, and expression-based cell cycle phase. Source data are provided as a Source data file.

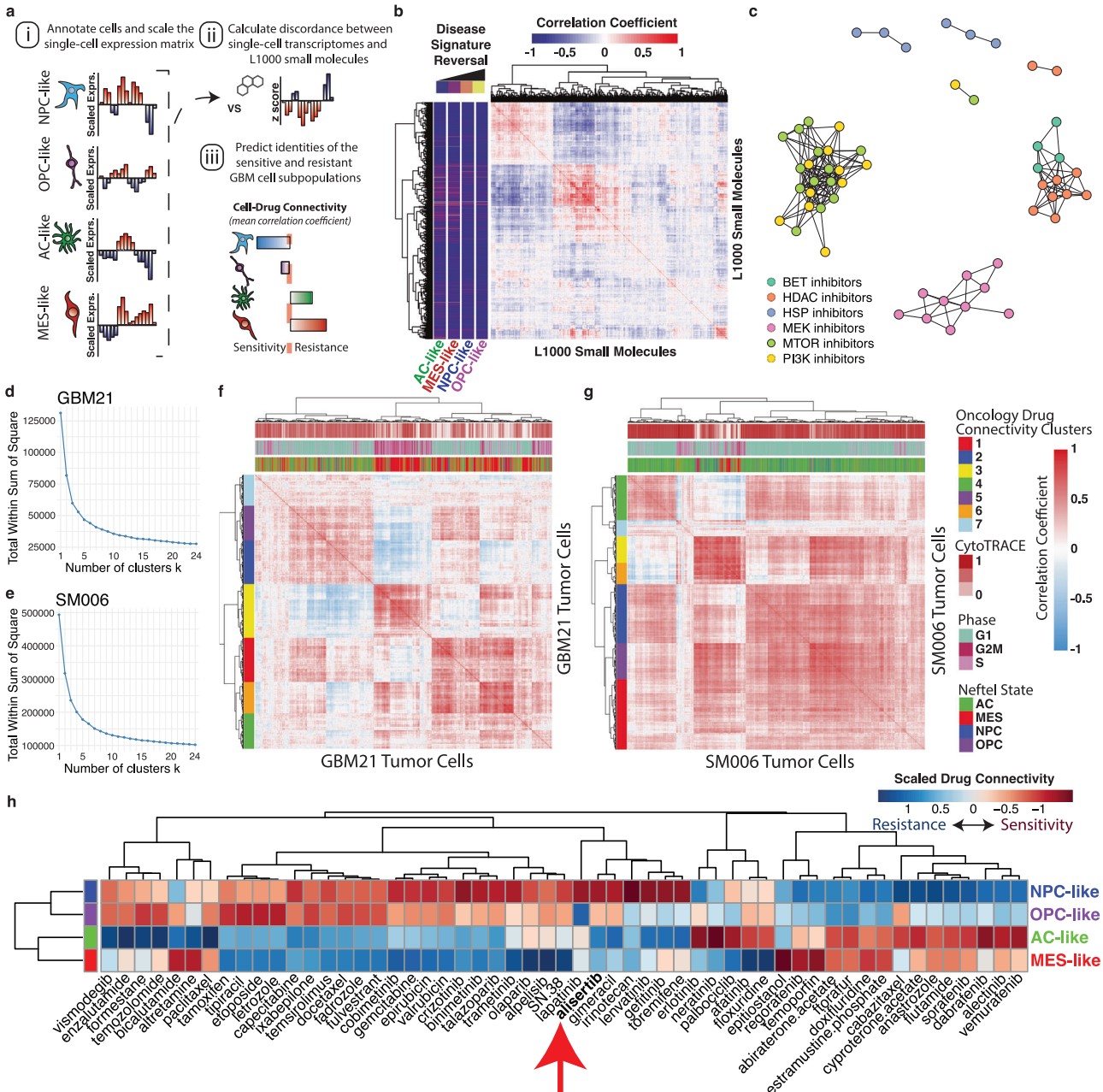

**Fig. 2 | Integration of single-cell expression and small molecule L1000 TCS signatures permits clustering of both compounds and cells by reversal of GBM cell transcriptional state-specific disease signatures. a** Schematic of single-cell sensitivity and resistance scoring. **b** Correlation matrix depicting similarities of L1000 small molecule TCSs by their connectivity to all individual cells within our single-cell atlas. Row annotations depict compound TCS discordance ratios for the reversal of AC-, MES-, NPC-, and OPC-like disease signatures calculated against non-neoplastic cells in the dataset. **c** Network plot of select L1000 small molecules colored by mechanism of action. Connections indicate a Pearson's $\rho > 0.7$ between small molecules by their calculated discordances against all single-cells in the GBM dataset as represented in (**b**). **d** Elbow plot depicting the within-cluster sum of squares by number of clusters k for GBM tumor cells from patient sample GBM21. **e** Elbow plot depicting the within-cluster sum of squares by number of clusters k for

GBM tumor cells from patient sample SM006 (Johnson[17] et al. dataset) **f**–**g**. Correlation matrices depicting pairwise Spearman correlations of single GBM tumor cells from individual patient tumor samples (GBM21, in-house) (**f**), SM006 (Johnson et al.)[17] (**g**) by their connectivity values to 63 FDA-approved oncology drug TCSs. Row annotation bars depict hierarchical clustering identities ($k=7$). Column annotations depict CytoTRACE scores, cell cycle phase, and assigned Neftel et al. state identity. **h** Heatmap of scaled pseudobulk single-cell drug connectivities for FDA-approved oncology compounds and alisertib that were significantly different between Neftel et al. cell states (Significance was determined by an empirical Bayes moderated two-sided $t$-test (limma lmFit, eBayes), and $p$-values were Bonferroni-adjusted). Heatmap color indicates predicted cell state sensitivity to each compound (red = relative sensitivity, blue = relative resistance). Source data are provided as a Source data file.

opposite direction of the disease signature expression, over the number of genes perturbed by compound TCS in the same direction as in the disease signature (Fig. 2b row annotations). Although a molecule can strongly reverse a bulk expression-based disease signature, reversal may be unique to a subpopulation of tumor cells. Many small

molecules are predicted to reverse the disease signatures of MES-like, AC-like, NPC-like, or OPC-like cells, but rarely are they predicted to strongly reverse the disease signatures of all transcriptional states (Fig. 2b row annotations, and Supplementary Data 4). Utilizing the same FDA-approved oncology drugs used in tumor cell clustering, we

find that many are differentially connected between the four transcriptional states (Fig. 2h).

## In silico perturbation analysis using scFOCAL predicts the intratumor response to alisertib treatment in vivo

Utilizing our calculated cell-drug connectivity data, we find that GBM cell transcriptional states show differing mean connectivities to different clinical oncology drugs (Fig. 2h). Here, we find 54 out of 64 compounds to have significantly different connectivities between cell states, determined by generalized linear model analysis using limma (BH-adjusted $p$-value < 0.05, Fig. 2h), including the aurora kinase inhibitor alisertib. Prior studies reveal that alisertib increases survival in an orthotopic xenograft mouse model of GBM. However, tumors eventually grow back, suggesting an acquired mechanism of resistance[13,22–24]. Aside from their role in cell cycle progression, the aurora kinases also play an important role in neurodevelopment and have been extensively studied as targets not only in GBM but many other cancers[15,24–50]. For this reason, we seek to identify whether a specific cell state confers resistance to aurora kinase inhibition in GBM. Overall, the alisertib TCS consists of 139 genes, 54 that are upregulated relative to vehicle control, and 85 that are downregulated. Of these 139 genes, 49 were differentially expressed among AC-, MES-, NPC-, and OPC-like cells. Utilizing differential expression between cells belonging to each transcriptional state, we find that the alisertib TCS predicts that alisertib treatment promotes the expression of MES-like specific genes such as *HMOX1* or *SQSTM1*, while reducing the expression of NPC-like specific genes such as *CDK4* or *EZH2* (Supplementary Fig. 4b). Additionally, the alisertib TCS discordance ratio, as utilized in SynergySeq, is highest against the NPC-like pseudo-bulk disease signature (Supplementary Fig. 4c). Using the calculated discordances of the alisertib TCS with the individual cells within the dataset, we performed an in silico drug connectivity analysis, where cells are predicted to be sensitive or resistant based on their expression concordance with the alisertib TCS (Fig. 3a, b). Predicted alisertib sensitive and resistant cell populations span source dataset, new diagnoses and recurrences, individual patient tumors, and cell cycle phases (Supplementary Fig. 5a–d). Labeling cells by their predominant meta-module enrichment, the predicted resistant cell populations across individual patients consistently show an increase in proportion of cells within MES-like states and depletion of NPC-like cells (Fig. 3c). The proportion shifts between NPC-like and MES-like states show the strongest differences, suggesting an NPC-like to MES-like predicted shift (Supplementary Fig. 4d). Differential expression testing between predicted alisertib-resistant and sensitive populations reveals a predicted resistance signature that was most strongly enriched in MES-like cells (Supplementary Fig. 5f). Further, expression of the NPC-like marker *CDK4* is significantly decreased in the predicted resistant cell population (Supplementary Fig. 5e).

To test the predictions from our in silico drug connectivity analysis using the patient tumor dataset, we use scRNAseq of orthotopic xenografts treated with alisertib or vehicle ($n = 3$ per group) to assess alisertib-induced shifts in transcriptional state composition (Fig. 3d). As most tumor cells in our patient dataset were AC-like (Fig. 1e–g), we utilized GBM22 PDX cells, which are molecularly classical, for our in vivo studies. Single cell analysis of GBM22 tumors isolated from mice reveals the presence of tumor cells within each of the transcriptional GBM cell states (AC-, MES-, NPC-, and OPC-like) (Fig. 3g, h, and Supplementary Fig. 6a–c). Importantly, in vivo alisertib treatment induces a depletion of NPC-like expression enrichment and an increase of MES-like enrichment (Fig. 3i, j), which aligns with our predictions (Fig. 3c, Supplementary Fig. 4d). Likewise, relative to more modest changes in proportion for AC-like and OPC-like cells, the proportion of GBM22 cells in a predominantly NPC-like state is reduced, while the proportion of GBM22 cells in a predominantly MES-like state is increased (Fig. 3i, j).

## In silico drug connectivity analysis predicts the intratumor compound discordance shifts observed in vivo

To assess whether resistant cell states predicted through in silico drug connectivity analysis of the patient tumor dataset represented those observed to persist through alisertib treatment in vivo, we directly compared the differences in global L1000 drug discordance between the predicted alisertib-sensitive and resistant cells in the patient data to those observed through comparing alisertib and DMSO-treated xenografts (Fig. 3k, and Supplementary Fig. 7a–h). We find that predictions of discordance shift from in silico drug connectivity analysis with alisertib in patient cells are predictive for certain classes of small molecules, including nucleoside reverse transcriptase inhibitors (Spearman's $\rho = 0.89$, $p = 0.033$), tubulin polymerization inhibitors (Spearman's $\rho = 0.82$, $p = 0.0068$), topoisomerase inhibitors (Spearman's $\rho = 0.82$, $p = 0.00011$), serotonin receptor agonists (Spearman's $\rho = 0.82$, $p = 0.0068$), MEK inhibitors (Spearman's $\rho = 0.77$, $p = 0.021$), HCV inhibitors (Spearman's $\rho = 0.77$, $p = 0.021$), HMGCR inhibitors (Spearman's $\rho = 0.83$, $p = 0.058$), and mTOR and PI3K inhibitors (Spearman's $\rho = 0.94$, $p = 0.017$) (Fig. 3k, and Supplementary Fig. 7a–h).

## Connectivity analysis with an L1000-derived panobinostat TCS separates vehicle-treated from panobinostat-treated cells in both neoplastic and myeloid cell populations

To assess the application of our prediction model in ex vivo settings, we utilized a publicly available scRNAseq dataset of GBM acute slice cultures treated with the pan-HDAC inhibitor panobinostat (Zhao et al., 2019) (Supplementary Fig. 8a, b)[51]. A panobinostat TCS is derived from the 2020 L1000 data release (Supplementary Fig. 8c) and used to score panobinostat connectivity of DMSO or panobinostat-treated neoplastic and myeloid cells. In both populations, panobinostat-treated cell populations showed significantly higher panobinostat connectivity values (Supplementary Fig. 8d, e). These findings are in agreement with those reported within the original manuscript in which this dataset was reported, wherein panobinostat elicits alterations to the transcriptional states of both tumor and myeloid cell populations[51].

## Connectivity analysis predicts drug combinations that synergize in vitro and in vivo

Having demonstrated the predictive power of our connectivity analysis framework to identify tumor cells resistant or sensitive to different perturbations, we seek to leverage our framework to identify synergistic combinations for glioblastoma (Fig. 4a). First, we utilized a recently published synergy screen performed across 24 independent glioma stem cell lines in spheroid culture to assess the ability of scFOCAL to predict relative synergy of small molecule combinations (Supplementary Fig. 9)[52]. Using erlotinib, lapatinib, pazopanib and sunitinib as reference compounds (Supplementary Fig. 9a), we ranked 13 small molecule combinations with these inhibitors using an scFOCAL based combination score (Supplementary Fig. 9b, top heatmap annotation). This score is representative of the specificity of a partner molecule in targeting the cell population resistant to the reference molecule and thus reflects the breadth of coverage a combination would have against the GBM cell transcriptional states globally represented within our patient GBM scRNAseq atlas. We find that this scFOCAL-based combination score strongly correlates with the mean observed BLISS synergy across cell lines of the combinations present within this screen (Spearman $\rho = 0.7$, $p = 0.01$), indicative of the potential that a representative GBM single-cell atlas can be used as a representative model for identifying more effective combinations (Supplementary Fig. 9c). Next, we focus on the OLIG2 inhibitor CT-179 due to its presumed ability to target OPC-like cells[53–57]. Using bulk RNA sequencing of molecularly classical GBM8 cells treated with 200 nM CT-179 or vehicle, we generate a dose-dependent CT-179 transcriptional signature comprising 841 differentially expressed genes for

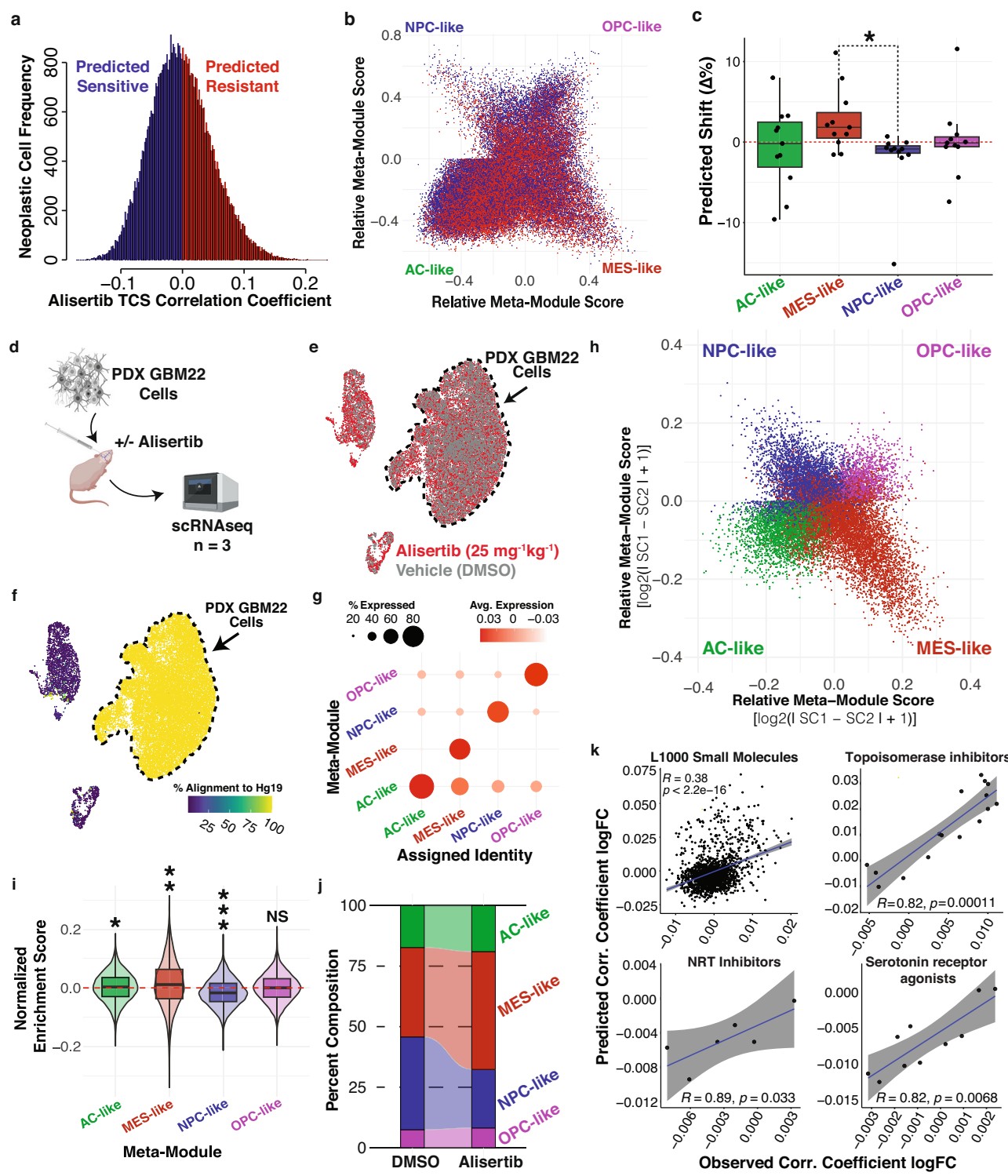

analysis of cell-CT-179 connectivity in our patient tumor atlas (Fig. 4b, and Supplementary Fig. 10a–c). By gene ontology analysis, the CT-179 response signature is enriched for processes including microtubule and tubulin binding, motor activity, and processes related to mitotic cell division (Supplementary Fig. 10b). Using the Spearman correlation of the directional CT-179 response signature with each tumor cell's scaled expression, cells are binned into predictive sensitive or resistant populations, which spanned all patients within our dataset (Fig. 4c). As expected, the CT-179 predicted sensitive population is primarily OPC-like (Fig. 4d), as evidenced by low CT-179 connectivity in OPC-like cells and high CT-179 connectivity in AC-like cells (Fig. 4e). Further, the

relative proportions of cells in an AC-like state is predicted to increase with CT-179 treatment, relative to the predicted depletion of cells in both NPC- and OPC-like states (Supplementary Fig. 10d). Using our established workflow for predicting the differential connectivities of L1000 small molecules between predicted sensitive and resistant cell populations, we identify 721 L1000 small molecules that exhibit a significantly reduced connectivity in CT-179 resistant cells compared to CT-179 sensitive cells (logFC < 0, adj.$p$ < 0.05) (Fig. 5a). As above, we also identified 706 discordant L1000 small molecules by their mean connectivity to the CT-179 resistant population (mean resistant cell connectivity, mRCC < 0) (Fig. 5b). 411 L1000 small molecules exhibited

**Fig. 3 | In silico perturbation of GBM tumor cell scRNAseq data using an L1000-derived alisertib TCS predicts an NPC-like to MES-like tumor response confirmed in vivo. a** Histogram of single-cell alisertib TCS connectivities. **b** Hierarchy plot of patient GBM tumor cells colored by predicted alisertib sensitivity ($\rho < 0$) or resistance ($\rho > 0$). **c** Box plot depicting per-patient difference in proportions of cells in each GBM cell transcriptional state in resistant vs. sensitive populations. Each dot represents the difference for an individual patient ($n = 11$). Boxplot elements represent the median (center line), quartiles (box limits), and 1.5x interquartile range (whiskers). Adjusted $p$-value from a two-sided Wilcoxon signed-rank test with Benjamini–Hochberg correction (MESvsNPC $p$.adj=0.019). **d** Schematic of in vivo experiments. (*Created in BioRender. Suter, R. (2025)* https://BioRender.com/3fhf7s6). **e** UMAP plot of pre-filter single-cell transcriptomes colored by treatment with alisertib or DMSO. **f** UMAP of captured cells colored by percent alignment to the human transcriptome (hg19). **g** Dot plot of pass-filter single-cell transcriptomes showing Neftel et al. signature expression, grouped by predominant transcriptional state module expression. **h** Two-dimensional hierarchical representation of GBM22 xenograft cells' relative enrichment scores for transcriptional state modules. Cells

are colored by assigned transcriptional state identity. **i** Violin and box plots quantifying enrichment shift of transcriptional state signatures in alisertib-treated xenograft cells ($n = 10,394$, cells from 3 pooled xenografts) normalized to DMSO-treated xenograft cell mean enrichments ($n = 5708$, cells from 3 pooled xenografts). Boxplot elements represent the median (center line), quartiles (box limits), and 1.5× interquartile range (whiskers). Violin width indicates the kernel density of the data. (Two-sided, one-sample Wilcoxon vs 0 with Benjamini–Hochberg correction: *$p$-adjusted=8e$^{-9}$; **$p$-adjusted=1.1e$^{-60}$;***$p$-adjusted=3.2e$^{-238}$; NS $p$-adjusted=0.79). **j** Alluvial plot depicting shift in relative proportion of transcriptional state identities in alisertib and DMSO vehicle control-treated xenografts. **k** Scatterplots of L1000 small molecules and subsets of compound classes' TCSs depicting log2FC predicted correlation shift vs. observed correlation shift in alisertib-treated xenografts. Differential small molecule correlations were calculated using limma. Spearman correlation coefficient R ($\rho$) and $p$ values were determined using a two-sided Spearman's correlation test (degrees of freedom=$n$-2). Shaded regions represent the 95% confidence interval of fitted regression lines. Source data are provided as a source data file.

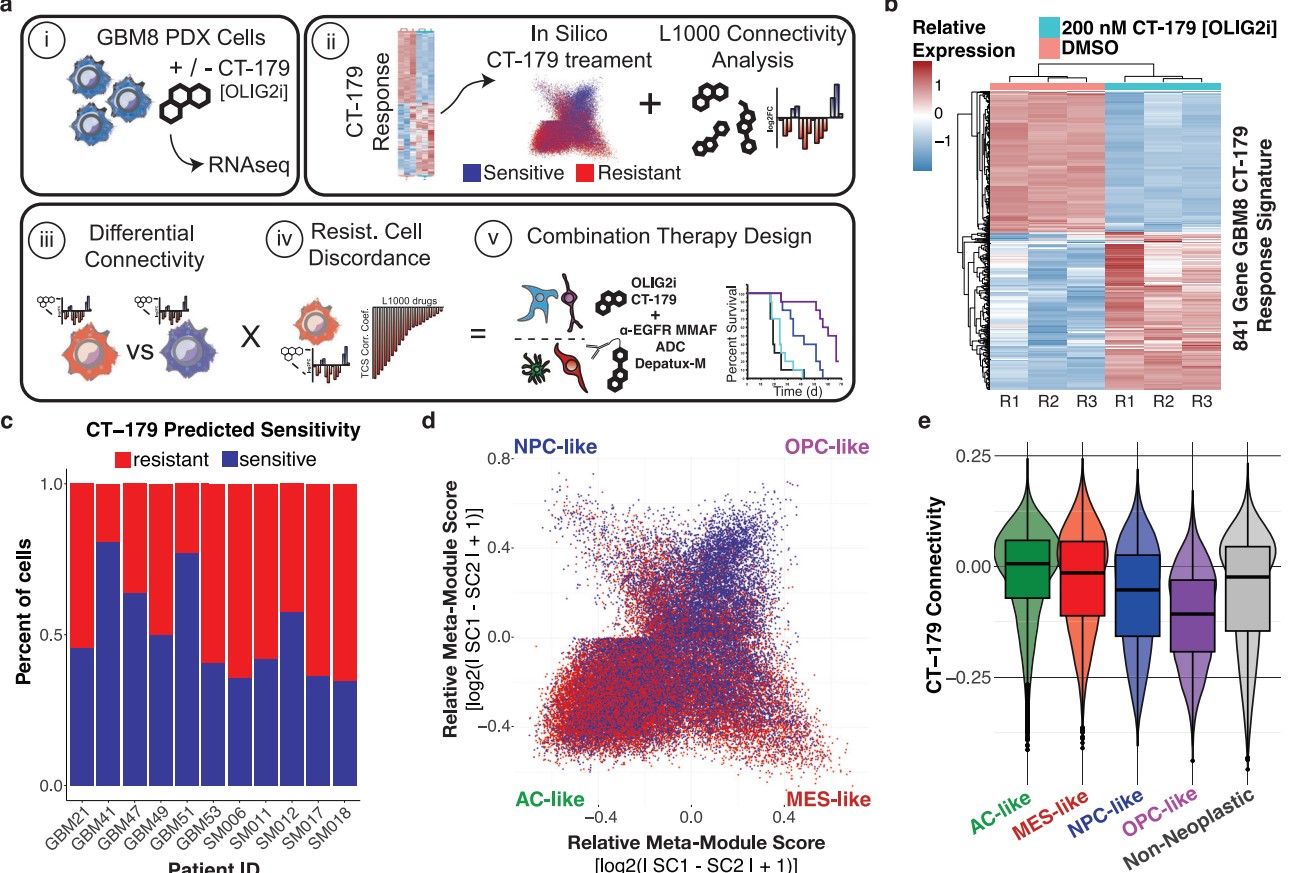

**Fig. 4 | The integration of a bulk-derived transcriptional response signature for the OLIG2 inhibitor CT-179 predicts targeting of OPC-like GBM cells. a** Diagram of workflow for identifying CT-179 combinations using scFOCAL and cell-drug connectivity. **b** Heatmap of GBM8 cells treated with vehicle or 200 nM CT-179 for 24 h. Columns represent biological replicates. **c** Bar plot depicting the proportion of tumor cells within each patient tumor predicted to be sensitive (CT-179 response $\rho < 0$) or resistant (CT-179 response $\rho > 0$) to CT-179 treatment. **d.** Hierarchy plot of GBM tumor cells arranged by their relative expression of Neftel et al. states colored by predicted sensitivity or resistance to CT-179 treatment. **e** Violin and box plot of

GBM cell CT-179 connectivity ($\rho$) grouped by assigned GBM cell transcriptional state (Kruskal–Wallis $p < 2.2 \times 10^{-16}$; For all pairwise comparisons, unpaired two-sided Wilcoxon signed-rank test with Benjamini–Hochberg correction $p$.adj $<2.2 \times 10^{-16}$). The center line of each box represents the median, box limits represent the first and third quartiles, and whiskers extend to the minimum and maximum data points. The violin outline indicates the kernel density of single-cell connectivity values. Data represents 53,542 single tumor cells aggregated across 11 individual patient samples. Source data are provided as a Source Data file.

both decreased connectivity and overall discordance with the CT-179 resistant population. By weighting the connectivity differentials of these molecules with their mean discordance to the CT-179 resistant cell population (mean Resistant Cell Connectivity, RCC), we

established a combination index score for each molecule within this intersect (Supplementary Data 6). Top hits by this metric included the EGFR inhibitor varlitinib, as well as docetaxel and indibulin, which elicit anti-neoplastic effects through alteration of tubulin and microtubule

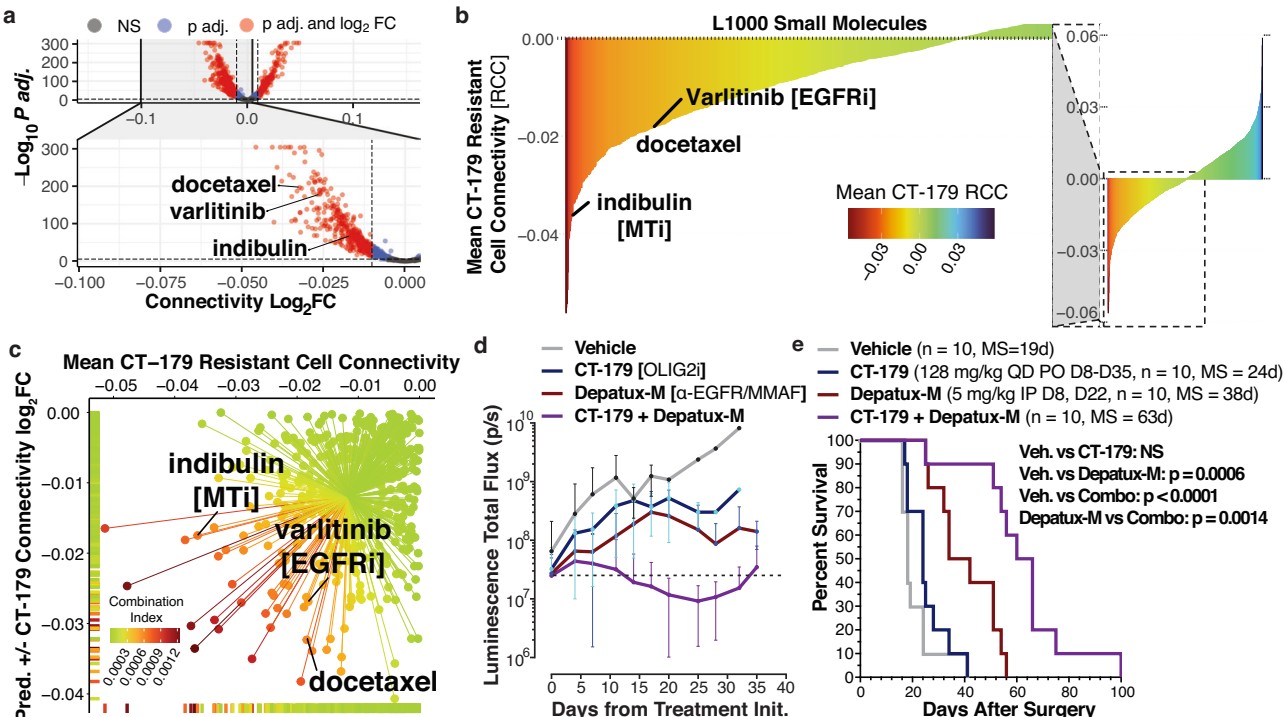

**Fig. 5 | An scFOCAL combination index predicts synergistic combination of an OLIG2 inhibitor CT-179 with Depatux-M (ABT-414), an anti-EGFR antibody MMAF drug conjugate. a** Volcano plot depicting results from limma-based differential drug connectivity between predicted CT-179 sensitive and resistant GBM cells. Patient ID was used as a covariate in the model design. **b** Bar plot of L1000 small molecule mean resistant cell connectivity (RCC) to predicted CT-179 resistant cells. Color depicts this same value. **c** Scatterplot of L1000 small molecules plotted by resistant vs. sensitive differential connectivity and mean CT-179 resistant cell connectivity. Colors depict the calculated scFOCAL combination index, the product of multiplying the differential connectivity log$_2$FC values by mean CT-179 resistant cell connectivity values for each individual molecule. Compounds highlighted in (**a**–**c**) are varlitinib, an EGFR inhibitor, and indibulin and docetaxel, which act through inhibition of tubulin. **d** Bioluminescence signal quantification of GBM6-eGFP-FLUC2 orthotopic xenograft tumors (means ± SD, Combo vs Vehicle: *p*.adj <0.005 from Day 14, Combo vs. CT-179 monotherapy: *p*.adj <0.05 from Day 11, Adjusted *p*-values from multiple *t*-tests (two-sided) with Holm–Šídák correction to control the family-wise error rate). **e** Kaplan–Meier survival curves of mice bearing GBM6-eGFP-FLUC2 orthotopic xenografts treated with indicated therapies (*n* = 10 per group). MS: median survival. *P*-values determined using Log-rank (Mantel-Cox) test. Source data are provided as a Source Data file.

assembly and dynamics (Fig. 5c)[58–61]. Indeed, *EGFR* and *OLIG2* are expressed by distinct GBM cells in AC-like and OPC- and NPC-like states, respectively (Supplementary Fig. 10d). These prioritizations lead us towards the use of the anti-EGFR monomethyl auristatin F (MMAF) antibody-drug conjugate Depatux-M (ABT-414), which would target *EGFR*-expressing cells with MMAF, a tubulin poison, in combination with CT-179. Mice bearing GBM6 orthotopic xenograft tumors were treated with vehicle, CT-179 alone, Depatux-M alone, or CT-179 and Depatux-M in combination. Treatment of mice bearing GBM6 patient-derived xenografts with the CT-179/Depatux-M combination increases survival over vehicle treatment or either monotherapy alone (Fig. 5e). In corroboration of the increased survival, bioluminescence imaging (BLI) reveals a significant reduction in tumor size following combination treatment with CT-179 and Depatux-M relative to CT-179 monotherapy alone (Fig. 5d). Collectively, these studies suggest that targeting OPC-like and AC-like cells simultaneously may be advantageous in vivo, and that maximizing drug discordance across the tumor cell landscape identifies more effective combinations.

### Release of the scFOCAL framework as an R package and a web application

We package the methodology defined and utilized here into an analytical framework termed scFOCAL (**S**ingle-**C**ell **F**ramework for -**O**mics **C**onnectivity and **A**nalysis via **L**1000), which we have made available as a Shiny web application and as an R package. As we have demonstrated, scFOCAL integrates bulk drug-response TCSs derived from the LINCS L1000 dataset with multi-subject scRNAseq data and facilitates the analysis of drug and cell discordance from multiple

perspectives. First, scFOCAL provides a simple approach to single-cell-derived disease signature generation and permits the analysis of disease signature heterogeneity across tumors relative to non-transformed cell types. Next, scFOCAL scores small molecules for their reversal potential of the generated disease signatures. Compounds predicted to reverse single-cell-derived disease signatures can then be utilized in scFOCAL's in silico drug connectivity approach. Here, as demonstrated with alisertib and panobinostat, L1000 TCSs are scored for discordance against the scaled expression of single-cells in a dataset. A custom transcriptional response signature can also be used, as we show in the case of CT-179. Users can then select a sensitivity cut-off (i.e., pseudo-dose), and the single-cell model will be split into predicted sensitive and resistant populations. Characteristics and visualizations, including proportions of annotated identities and predicted drug sensitivities, can then be explored. Additionally, users can identify predicted synergistic compounds through calculation of an scFOCAL combination index score for each L1000 compound with their drug of interest. All data generated by users of scFOCAL are available to be downloaded for ease of access. Lastly, the scFOCAL shiny application integrates seamlessly with the R package OrthologAL and the OrthologAL shiny app, which can prepare Seurat objects for dual-species xenografts and non-human expression data for analysis in scFOCAL[62]. Single-cell connectivity scores for all compounds against the patient GBM scRNAseq atlas presented in this study are downloadable using the scFOCAL Shiny web application, which can be found at https://robert-k-suter.shinyapps.io/scFOCAL alongside instructions for R package download at https://github.com/AyadLab/scFOCAL (Fig. 6).

## Discussion

The concepts of disease signature reversal and single-cell omics technologies are becoming a mainstay of drug repurposing and personalized medicine[11,63–65]. Using principles of disease signature reversal, we show that drug response signatures from the LINCS L1000 dataset can identify compounds and compound combinations effective at eliminating specific GBM transcriptional subtypes[12,13]. Based on these findings, we propose that if the transcriptional response signatures represent the expression of cells remaining beyond therapeutic pressure, then single cells with expression programs most concordant to that signature would show the highest probability of persistence after treatment and would therefore drive therapeutic resistance.

Here, we describe an analytical framework for the identification of small-molecule perturbagens that target distinct transcriptional niches of GBM tumor cells. We create a GBM scRNAseq atlas through the integration of a publicly available dataset with our scRNAseq dataset of newly diagnosed and recurrent patient GBM tumors (Fig. 1, and Supplementary Fig. 1). Integrating pharmacological transcriptional response signatures from the LINCS L1000 dataset into the atlas facilitates pharmacotranscriptomic analysis and leads to the development of an in silico drug connectivity analysis platform, scFOCAL, to predict the sensitive and resistant cell populations that would result from specific small molecule treatments (Fig. 3). Importantly, we find that GBM tumor cells can be classified distinctly through their connectivities to current FDA-approved and other clinically relevant oncology drugs (Fig. 2d–g). Clustering tumor cells within each patient on their connectivities to FDA-approved oncology drugs reveals patterns relating to Neftel et al. transcriptional state, cell cycle, and CytoTRACE (Fig. 2d–g, and Supplementary Fig. 3a, b), indicating that GBM tumor cells within different developmental and differentiation states possess unique vulnerabilities to different small molecule perturbations, independently from drug sensitivities related to cell cycle phase.

Investigating aurora kinase inhibition as a use-case, we use scFOCAL to predict the sensitivity of NPC-like cells, and the resistance of MES-like cells to treatment with alisertib (Fig. 3, and Supplementary Figs. 4 and 5). Utilizing single-cell RNA sequencing of orthotopic xenografts treated with alisertib or vehicle, we validate our predictions and demonstrate that MES-like expression and the proportion of predominantly MES-like cells increases with alisertib treatment, while NPC-like expression and the proportion of predominantly NPC-like cells decreases (Fig. 3, and Supplementary Fig. 6). scFOCAL, therefore, has the potential to predict tumor transcriptional shifts occurring under therapeutic pressure (Fig. 3k). Here, we also show scFOCAL's ability to predict shifts in connectivity to other drugs following alisertib treatment, demonstrating particular predictive power in the context of select compound classes including topoisomerase inhibitors, serotonin receptor agonists, and NRT (Nucleoside Reverse Transcriptase) inhibitors (Fig. 3k, and Supplementary Fig. 7). Interestingly, topoisomerase inhibition has been shown to downregulate *AURKA* expression, which may explain why most are predicted by scFOCAL and in vivo to be less discordant with alisertib-resistant GBM cells[66]. NRT inhibitors and serotonin receptor agonists may represent future interests in combination with alisertib, as we both predict via scFOCAL and observe in vivo that some of these compounds are more discordant with alisertib-resistant or alisertib-treated tumor cells.

In another use case, we utilize a publicly available single-cell dataset of GBM acute slice cultures treated with the HDAC inhibitor panobinostat to reinforce that L1000-derived TCS connectivity is predictive of treatment response at single-cell resolution (Supplementary Fig. 8). Importantly, we find that cell-panobinostat TCS connectivity predicts single-cell resolution alterations in both the tumor and myeloid cell populations of the TME, as reported in the original publication (Supplementary Fig. 8)[51]. These findings should be

explored further in the future to assess the potential of scFOCAL-based prioritization of small molecules targeting distinct TME cell phenotypes. Importantly, scFOCAL prediction of targeted and resistant GBM cell transcriptional states (NPC, OPC, AC, MES) using the panobinostat TCS correlates with the shifts in Neftel et al. state proportion following actual treatment of the slices (Supplementary Fig. 8).

Acquired treatment resistance is common in glioblastoma, which is notoriously heterogeneous and plastic at single-cell resolution[6–8,10,67–69]. Longitudinal studies in glioblastoma implicate epigenetic intratumor heterogeneity in standard-of-care treatment resistance[5,70–72]. Transcriptional subtype switching occurs often upon GBM recurrence[5,10,72]. Thus, the characterization of the transcriptional response of acquired therapy resistance and the identification of means by which to circumnavigate this resistance are essential. Importantly, others show that the most discordant cells within a tumor are predictive of patient outcome in the clinic and, further, that predictors of treatment response based on single-cell RNA sequencing data outperform those based on bulk expression profiles[63]. They also demonstrate that the predictive power of the data lies within the most 'resistant' cells in the dataset. Here, with confidence that scFOCAL accurately identifies sensitive and resistant transcriptional states of cells for specific treatments, we posit that identifying drugs, which target these resistant cells in the context of a reference treatment will enable the discovery of more effective combination therapies.

We evaluate this hypothesis by first developing a combination scoring index that weights reference-drug resistant cell discordance with the differential drug connectivity across reference-drug sensitive and resistant cell populations (Supplementary Fig. 9). We then leverage this same combination scoring approach on our patient scRNAseq atlas using a bulk RNA-seq-derived transcriptional response signature for the OLIG2 inhibitor, CT-179. CT-179 is a small molecule recently characterized as a potential therapeutic for other brain cancers, such as medulloblastoma[53–57]. OLIG2 has also been characterized as a key dependency in GBM, regulating a neurodevelopmental transcriptional program key in gliomagenesis and maintenance of different tumor cell states[73,74]. However, targeting of OLIG2 alone is not sufficient in the face of GBM intratumor heterogeneity. As expected, cell and CT-179 connectivity predict the targeting of OPC-like GBM cells (Fig. 4e) and an AC-like CT-179 resistant state. In line with this observation, combination indexing with scFOCAL highlights EGFR and tubulin inhibitors (varlitinib, indibulin, docetaxel) within the top 10% of combination scores from the 411 clinical compounds passing our connectivity-based filters. Leveraging this finding, we identify Depatux-M (ABT-414), an antibody-drug-conjugate consisting of an EGFR antibody conjugated to MMAF, a tubulin inhibitor, as a viable candidate for combination treatment with CT-179. While our analysis identifies other top hits, including the PI3K inhibitor taselisib and the PLK1 inhibitor volasertib, we prioritize the combination with Depatux-M. This is due to Depatux-M's mechanistic overlap with multiple compounds identified by our in silico screen, and its established clinical profile, including brain penetrance, a known safety record, and advancement to Phase III GBM trials[75]. In support of this, Depatux-M synergizes with CT-179 to attenuate tumor growth and to extend survival in vivo (Fig. 5d, e)[76].

Although we develop a computational platform to predict drug combinations based on scRNAseq information, we acknowledge limitations to our work. scFOCAL, as well as any other predictive platform, is reliant on the data upon which it is built. For one, most of the input for scFOCAL is from LINCS, and therefore, compounds must be profiled using the L1000 assay. To temper this limitation, we include a feature to allow investigators to add their own TCSs from RNA-seq data of treated cells, which should expand the utility of scFOCAL. Additionally, scFOCAL uses scRNAseq data as its input, and both cell types (e.g., neurons) and spatial architecture are lost during the processing of samples. Therefore, the integration of single-cell gene expression with other diverse data types, such as DNA methylation, epigenetic

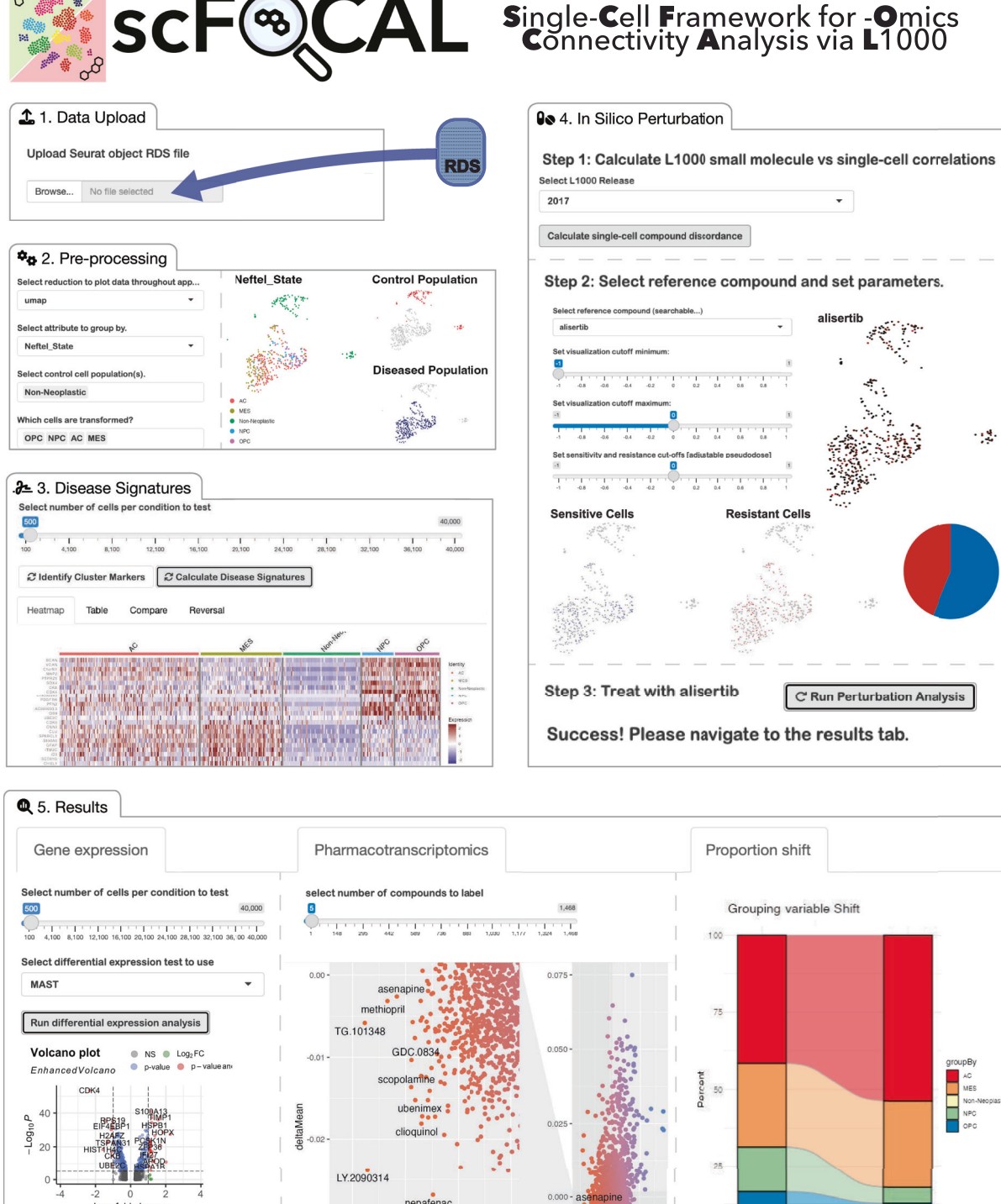

profiling of histone acetylation, proteomics, and spatial transcriptomics, could enhance and refine the predictive power of scFOCAL into a more comprehensive prediction model. Similarly, it may be possible to use different reference normal cells to enhance concordance and discordance predictions for tumor cells in scFOCAL with the addition of the aforementioned data. Exploring the utility of single-nuclei RNA-seq (snRNAseq) datasets, for example, may provide both additional non-tumor cells and different tumor cell states not captured by scRNAseq. Future studies are required to compare scRNA and snRNAseq predictions from scFOCAL and evaluate the capacity to integrate the data they provide.

Importantly, the response signatures themselves used to calculate compound discordance with single cells could be optimized further. Many alternate approaches to TCS generation have been validated[77,78].

**Fig. 6 | The scFOCAL framework is available for use as a shiny web application or an R package.** Screenshots depicting the scFOCAL graphical user interface workflow. **1. Data Upload:** Users may upload a pre-processed and annotated Seurat object .RDS file as input for analysis. **2. Pre-processing:** Users may select their desired dimensionality reduction (i.e., PCA, UMAP, tSNE) to use for visualization, the identities of non-tumor cell populations, and the identities of the different tumor cell populations within their dataset. **3. Disease Signatures:** Users may calculate pseudobulk disease signatures for each individual tumor cell population relative to the non-tumor control cell populations, and calculate discordance ratios for small molecules against each calculated signature at a population or individual subject (i.e., patient) level. **4. In Silico Perturbation:** Users may select the release year for L1000-derived transcriptional consensus signatures (TCSs) to use, and initialize cell-by-cell cell-drug connectivity scoring. Once scoring is complete, users may select a molecule of interest or upload a signature file for an external perturbation, visualize it's connectivity across the dataset, select sensitivity and resistance thresholds, and intialize scFOCAL in silico perturbation analysis. **5. Results:** Once in silico perturbation analysis has been initialized, users may perform differential expression to compare predicted sensitive and resistant tumor cell populations, visualize predicted shifts in proportions of selected annotations, and initialize scFOCAL combination indexing analysis to predict synergistic combinations.

Additionally, gene expression altered over longer periods compared to the time points assessed within the L1000 dataset (e.g., 24 h) may be more representative of long-term alterations that would occur in situ in GBM as well as other diseases. The selection of the most tissue-relevant cell lines from among those profiled within the L1000 dataset may also improve the predictive power for certain tissues. Although we use all perturbed L1000 lines as opposed to GBM lines only, we have demonstrated in a prior publication the general applicability of L1000 TCSs derived from non-GBM cell lines to predict efficacy in GBM and other brain cancer types[12,13,19]. Lastly, in the context of combination design, scFOCAL effectively ranks compounds based on their ability to maximize cell discordance across the heterogeneous GBM tumor transcriptional landscape; however, this information must be combined with other factors of consideration, such as GBM tumor biology, blood-brain barrier penetrance, and toxicity.

While other tools exist for exploring and utilizing the L1000 CMap datasets, such as the L1000CDS2 or TargetTranslator, scFOCAL is a tool with distinct functionality and uses[78,79]. The L1000CDS2 serves as a browser for identifying single datapoints in the data correlated with input gene signatures, but does not facilitate sample-by-sample analysis, integration with standard single-cell workflows, nor an implementable combination prioritization strategy[78]. TargetTranslator is a powerful framework for identifying druggable targets using gene expression data, but it does not inform on what the resistant cell states to inhibiting these targets would be, or which targets should be engaged simultaneously[79]. Importantly, scFOCAL permits analyses utilizing L1000-derived TCSs as well as custom input response signatures, a feature that can be used to characterize novel drugs in the context of similarity to other L1000 molecules, predicted target cell populations, and prioritization of additional combinations.

Collectively, our studies demonstrate that the application of disease signature reversal to the transcriptional profiles of single cells can predict sensitive and resistant cell populations within GBM tumors, aiding in the selection of therapeutic compound combinations for the treatment of patients with GBM. This highlights the potential utility of single-cell transcriptomics for future clinical trial design, wherein patients could be stratified based on the abundance of specific cell populations within tumors. Further, scFOCAL aligns with dynamic precision medicine initiatives, where intratumor dynamics in response to treatments can be modeled and predicted to inform combinatorial or sequential therapeutic strategies that effectively target an evolving tumor cell landscape[80]. Lastly, the scFOCAL platform could be leveraged to investigate the potential to reverse distinct cellular phenotypes to more desirable ones in other disease types. All analyses outlined are easily accessible through the scFOCAL R package and application[16]. scFOCAL includes an easy-to-use graphical user interface developed on the R Shiny platform, facilitating its use on pre-processed datasets for users without coding backgrounds, which increases accessibility to a broader range of pharmacology domain experts and represents a significant advance towards data-driven personalized medicine.

## Methods

### Collection of newly diagnosed and recurrent GBM tumors

Six (6) patients with GBM permitted tissue collection before tumor resection under approved IRB study 20170887 (MODCR00002196). Of the 6 enrolled patients, 3 were newly diagnosed with GBM and 3 had tumor recurrence after previously receiving standard of care treatment with temozolomide chemotherapy and radiotherapy. The 3 newly diagnosed patients had not received any treatment before resection. After surgical resection, University of Miami Hospital pathologists separated the resected tissue, which was then transferred from the operating suite to the laboratory on ice. Tissue from the enhancing edge was then used for dissociation and single-cell RNA sequencing (see below). Patient samples were obtained with written, informed consent.

### GBM cell xenograft models

Human patient-derived GBM cells (GBM6, GBM8, GBM22) were obtained from the Mayo Clinic Brain Tumor Patient-Derived Xenograft National Resource. GBM22 cells were transduced to express GFP (pSMAL-CellTag-V1), while GBM6 cells were transduced to express eGFP-FLUC2 as previously described[76,81–83]. Briefly, lentiviral particles were packaged in HEK293T cells, and transduction to primary GBM22 and GBM6 cells was performed in the presence of 5 μg/ml polybrene. GBM8 was derived from a newly diagnosed tumor in a female patient and is molecularly IDHwt, Classical, EGFR amplified, and MGMT methylated. GBM22 was derived from a newly diagnosed tumor in a male patient and is molecularly IDHwt, Classical, MGMT methylated. GBM6 was derived from a newly diagnosed tumor in a male patient and is molecularly IDHwt, with EGFRVIII mutation, MGMT unmethylated.

### Mouse models

All research was conducted in compliance with ethical regulations. Athymic nude mice (Crl:NU(NCr)-Foxn1[nu]) from Charles River Laboratories (stock 553) were utilized for in vivo experiments. Orthotopic implantations took place when mice were 6–8 weeks old. All procedures were approved by the University of Miami IACUC, protocol number 18-014, or by the Mayo Clinic IACUC, protocol number A4595-19. All animal procedures were performed per guidelines and regulations, including ARRIVE guidelines[13,76,84,85]. Mice were housed in the respective Department of Veterinary Resources on corn cob bedding in individual HEPA ventilated cages (Innocage® IVC, Innovive USA) on a 12-h light-dark cycle at 68–79 °F (20–26.1 °C) and 30–70% humidity. Animals were fed water (reverse osmosis, 2 ppm Cl2) and an irradiated rodent diet (LabDiet5053 PicoLab Rodent Diet 20) ad libitum. Animals were monitored for signs of lethargy or head-tilting. For IVIS studies, maximum tumor volume did not exceed 60 mm$^3$, in accordance with the approved protocol. Experimental groups for these experiments were assigned using a random number generator.

## Preparation of newly diagnosed and recurrent GBM tumors for scRNAseq

We performed single-cell RNA sequencing on the 6 patient GBM tissue samples collected at the time of surgical resection ($n = 3$ newly diagnosed, $n = 3$ recurrent, Fig. 1) After resection, University of Miami hospital pathologists separated the GBM tissue, which was transferred from the operating suite to the laboratory on ice. Tumor samples were dissociated using the Worthington Biochemical Papain Dissociation System[86]. Tumor tissue was finely minced with a sterile surgical blade and incubated in sterile Earl's Balanced Salt Solution (EBSS) containing 20 units of papain per mL in 1 mM L-cysteine with 0.5 mM EDTA and DNase for 45 min at 37 °C while shaking. Following papain incubation, the tissue was gently triturated with a 5 mL serological pipette until it was adequately dissociated. The resulting suspensions were pelleted and resuspended in EBSS with ovomucoid protease inhibitor, bovine serum albumin (BSA), and DNase. This suspension was layered on top of ovomucoid protease inhibitor solution with BSA to form a gradient, which was centrifuged at $70 \times g$ for 6 min. The supernatant, which contained cellular debris, was removed. The pellets were resuspended, and the gradient centrifugation was repeated. The resulting pellet was then washed 2 times with ice cold PBS with 0.1% BSA, passed through a 0.45 µM cell strainer, and resuspended in ice-cold PBS with 0.1% BSA. This resulting suspension was then checked for cell viability by staining an aliquot with acridine orange and propidium iodide (AO/PI) and counted on a Nexcelom K2 Cellometer. All samples had cell viability greater than 90% prior to loading of the single-cell chips. Single cells were captured using a 10X Genomics Chromium Controller, and patient single-cell gene expression libraries were prepared using 10X Genomics 5′ Gene Expression chemistry. All libraries were sequenced on an Illumina NextSeq500.

## Pre-processing and quality control of patient-derived scRNAseq data

Raw scRNAseq dataset was aligned and demultiplexed using Cell-Ranger (10X Genomics), and the resulting filtered unique molecular identifier (UMI) matrices were used for downstream analysis in the R package Seurat (v4)[87–91]. Within Seurat, poor quality cells were further filtered by removing outliers based on the number of detected UMIs and unique transcripts, as well as percent expression of mitochondrial transcripts. Pass-filter cells were those with greater than 200 and less than 4500 detected features, and with less than 12.5% mitochondrial expression. Data from 42,839 pass-filter single-cell transcriptomes across 33,469 features were normalized and variance stabilized using a regularized negative binomial regression implemented within the R package scTransform, and individual patient datasets were integrated using an anchor-based technique utilizing the Pearson residuals[87–91].

## Harmonization and integration of in-house and external scRNAseq datasets

Previously published single-cell RNA sequencing data from Johnson et al. (2021) were downloaded from Synapse (https://synapse.org/singlecellglioma)[17]. The five IDHwt GBM samples out of the eleven total glioma samples available in the dataset were used[17]. To integrate the Johnson et al. dataset with our internal dataset, each was first split into individual datasets based on the patient source. Each individual patient's expression matrix was then independently prepared for integration using scTransform, and integrated together using the anchor-based technique in Seurat (Fig. 1b). 28,130 single-cell transcriptomes from Johnson et al. were integrated with our dataset for a total of 70,969 single cells. Integrated expression values were used for initial shared-nearest-neighbor (SNN) clustering, dimensionality reduction, and cell type identification (Fig. 1a). Downstream analysis, including differential expression and enrichment scoring, was performed using log-normalized counts.

## Generation of GBM cell transcriptional state-specific disease signatures

SNN clustering on the patient-derived, variance-stabilized, integrated scRNAseq data identified distinct clusters of neoplastic and non-neoplastic cells. To confirm the identification of these neoplastic cells, we assessed transcriptional diversity with the R package CytoTRACE using raw counts as input. CytoTRACE scores cells based on their developmental potential, reflecting the plasticity of GBM tumor cells relative to the TME (Supplementary Fig. 1)[18]. Non-neoplastic cell identities were assigned by expression of canonical cell type markers identified through differential expression with MAST (Model-based Analysis of Single-cell Transcriptomics) (Supplementary Fig. 1, and Supplementary Data 1)[92]. The neoplastic cells were further subdivided and assigned GBM cell state identities using GBM cell transcriptional state signatures previously reported in Neftel et al. (2019) and the R package singscore (Supplementary Fig. 1)[6,93]. Briefly, the raw count matrices across all samples were log-normalized and scaled, regressing out S-Phase and G2/M-Phase expression module scores as calculated in Kowalczyk & Tirosh et al. (2015). These scaled data were then used to generate gene rank lists for each individual cell. Using the R package *Singscore*, gene set enrichment analysis was performed on each cell independently for the six Neftel et al. GBM cell state signatures (meta-modules). The means of NPC1- and NPC2-like, and MES1- and MES2-like state enrichment scores were calculated and represented as NPC-like and MES-like enrichment, respectively. Each cell's assigned transcriptional identity reflects its predominant, most-enriched GBM cell state expression program (AC, MES, NPC, OPC). Thus, the identities assigned within this manuscript indicate the predominant transcriptional program of that cell. Hierarchy plots were generated using the R package scrabble[6]. Differential expression testing using MAST was performed between each of the state-specific clusters and all non-neoplastic cells to generate disease signatures (Fig. 2a, and Supplementary Data 2)[92]. MAST was used to account for the technical pitfall of stochastic dropout and the resulting bimodal expression distributions inherent in scRNAseq. Individual patient signatures were generated by comparing each individual's tumor cells to all non-neoplastic cells from all patients within the dataset (Supplementary Data 3). Heatmaps were generated using the R package pheatmap[94].

## In silico drug connectivity analysis with the aurora kinase inhibitor, alisertib

The overlap of the alisertib TCS and results from differential expression testing using MAST between patient neoplastic cells grouped by predominant transcriptional state enrichment was calculated to identify potential state-specific expression perturbed by alisertib (Fig. 4a). Single-cell Spearman' $\rho$ values for the alisertib TCS were extracted from the L1000 small molecules versus single-cell correlation matrix. Neoplastic patient cells were divided into predicted alisertib resistant and alisertib sensitive populations wherein cells concordant (Spearman's $\rho > 0$) with the alisertib TCS were identified as a predicted resistant population, and those discordant (Spearman's $\rho < 0$) with the alisertib TCS were identified as a predicted sensitive population (Fig. 3a, b). The predicted shift in cell states was then calculated, demonstrating an increase in MES-like cells and a decrease in NPC-like cells (Fig. 3c). Proportions across the source dataset, initial occurrence or recurrence, source patient, and cell cycle phase were also compared (Supplementary Fig. 4a–d). Proportion shifts of cells predominantly in each transcriptional state within each individual were compared using ANOVA with a Games-Howell post-hoc test. Resulting adjusted $p$-values were used to visualize the relative probability of predicted directional state abundance shifts (Supplementary Fig. 4d). Differential expression testing between predicted resistant and sensitive populations was performed using MAST, and cells were scored for their enrichment of overexpressed genes using Seurat's AddModule-Score function (Supplementary Fig. 5f).

## GBM cell orthotopic xenograft, alisertib treatment, and scRNAseq library creation

Human patient-derived GBM cells (GBM22) expressing GFP obtained from the Mayo Clinic Brain Tumor Patient-Derived Xenograft National Resource were implanted in the brains of nude mice (Fig. 3d)[22,76,85]. Cells were visualized to confirm GFP expression prior to intracranial implantation into mice. Briefly, 6–8 weeks old Nu/Nu mice (Charles River Laboratory) were anesthetized with ketamine (100 mg/kg) and xylazine (10 mg/kg), and a hole was drilled through the skull (1 mm anterior and 2 mm lateral to bregma). Using a Hamilton syringe, 150,000 GBM22 cells suspended in PBS were injected stereotactically at a depth of 2 mm. The skin incision was closed using surgical glue with sutures or wound clips and mice were given buprenorphine as analgesic. Procedures were approved by the University of Miami IACUC, protocol number 18-014. All procedures were performed in accordance with guidelines and regulations, including ARRIVE guidelines.

Following tumor implantation, mice recovered for a period of 7 days. GBM22 orthotopic tumor-bearing mice were treated intraperitoneally with either alisertib (25 mg/kg) (drug formulation: 5% DMSO/30% PEG400/5% Tween-80/60% PBS) or vehicle control 5 days per week and monitored daily for signs of cognitive decline and body weight (BW). Tumors were collected at the time of initial signs of cognitive decline or 15% of BW loss.

Tumor cells from three individual mice within each treatment group were captured and pooled for single cell RNA-sequencing. Tumor-bearing mice were perfused with PBS, and their brains were removed. Tumor was isolated and dissociated using the Worthington Biochem Papain dissociation kit. Cells were washed two times using ice-cold PBS with 0.1% BSA (w/v), then checked for viability using AO/PI dye and a Nexcelom K2 Cellometer. Cells from the dissociation suspension were loaded into a 10X Genomics' Chromium chip using 10X Genomics' NextGEM 3′ chemistry targeting the capture of ~10,000 cells per sample. With each individual animal experiment, paired-end library sequencing of DMSO and alisertib-treated single-cell libraries was performed in tandem on a single S1 NovaSeq flow cell, with asymmetric read lengths as recommended by 10X Genomics.

## Pre-processing and quality control of in vivo scRNAseq data

Sequencing basecalls were input into CellRanger V3.0.2 for demultiplexing and fastq generation. Alignment and gene quantification were first performed in tandem to a dual-species reference transcriptome for both hg19 (human) and mm10 (mouse), obtained from 10X Genomics (refdata-cellranger-hg19-and-mm10-3.0.0.). Using the R package Seurat (V4.0.1), the percentage of raw UMI counts within each cell aligning to either mm10 or hg19 transcriptomes was calculated and used in tandem with SNN clustering to separate out human GBM22 cells from contaminating mouse cells (Fig. 3e). GBM22 cell barcodes were then used to isolate these human cells from the data aligned to the GrCH38 (human) transcriptome for downstream gene expression analysis (Fig. 3f). Outlier cells were removed based on UMI count, detected gene count, and percent of mitochondrial transcript counts. Cells with less than 200 or more than 6000 detected genes, or with more than 10% of mitochondrial genome alignment were removed for downstream analysis. Pass-filter data was comprised of 16,102 GBM22 single-cell transcriptomes with 10,394 measured features.

## Generation of a CT-179 GBM cell transcriptional response signature

GBM8 cells (Primary, IDHwt, Female, Classical, EGFR amp, MGMT methylated) were obtained from the Mayo Clinic Brain Tumor Patient-Derived Xenograft National Resource. GBM8 cells were treated with 100 nM CT-179, 200 nM CT-179, or vehicle control, and RNA was extracted following a 24 h treatment period. RNA sequencing was performed on the NovaSeq 6000. FastQ files were aligned to the hg19

human genome to determine raw gene counts[95]. Differential expression between 200 nM CT-179 and vehicle-treated GBM8 cells was performed using DESeq2 (Supplementary Fig. 10a). Differentially expressed genes retained within the signature showed a logFC greater than 0.425, calculated as two standard deviations from the mean of all logFC values, and an adjusted $p$-value less than 0.05 (Fig. 4b). Gene ontology analysis was performed on this filtered signature using clusterProfiler (Supplementary Fig. 10b)[96–100]. Enrichment of the signature in 100 nM CT-179 treated, 200 nM CT-179 treated, or vehicle-treated GBM8 samples was calculated using the R package Singscore and compared to confirm the dose-dependence of our signature (Supplementary Fig. 10c).

## In vivo evaluation of CT-179 and Depatux-M in combination

Orthotopic implantation of GBM6-eGFP-FLUC2 was performed using a previously established protocol[76]. Mice were anesthetized with ketamine and xylazine (100 mg/kg and 10 mg/kg, respectively), then placed in the Stoeling #51615 frame for sterotactic guidance. The incision site was disinfected with Nolvasan and the skull was exposed. Using a Dremel with #8 bit, the skull was drilled at a point 1 mm anterior and 2 mm lateral to bregma. 150,000 cells (in a suspension of 50,000 cells/mL) were injected sterotaxically over 1 min using a 26G Hamilton syringe at a depth of 2 mm. One week following surgery, mice were treated with vehicle, CT-179, Depatux-M, or a combination of CT-179 and Depatux-M ($n = 10$ per group). CT-179 was formulated in 0.5% methylcellulose, 0.5% Tween80 in water, and stored at 4 °C. Fresh batches were made on the first day of dosing, and on day 18 of dosing. CT-179 formulation was stirred at RT for 30 min prior to use, and was delivered at a dose of 128 mg/kg QD PO from days 8 to 35. Depatux-M (ABT-414) was formulated in PBS, and was delivered at a dose of 5 mg/kg IP on day 8 and day 22. Tumor formation was confirmed, and tumor volume was assessed using BLI (Fig. 5d). BLI was performed using an IVIS 2.50 cooled charged-coupled device camera (Xenogen 200 series) and Living Image software[76]. Mice were dosed with 10 mg/kg Cycluc1 by intraperitoneal injection and imaged 10 min later under isoflurane anesthesia[76]. A moribund state was defined by outward neurological deficits (altered gait, hunched posture, circling, poor body condition score). Survival differences were assessed using the Log-Rank test and Kaplan–Meier analysis (Fig. 5e).

### Quantification and statistical analysis

**Ranking of L1000 compounds against GBM cell transcriptional state disease signatures.** The L1000 assay contains 978 landmark genes that can be measured to algorithmically infer the expression of 11,300 additional genes[77,78,101]. Using a previously published algorithm to rank discordance of L1000 perturbations, small molecules' response signature discordance with scRNAseq-derived disease signatures were calculated for each compound in the dataset[13,14,77,101]. This disease-specific discordance score was calculated for all compounds within the L1000 library as a ratio of the number of genes with discordant expression (opposite gene expression direction/sign) between the TCS and the cell state disease signature over the number of genes with concordant expression (same gene expression direction/sign) between the TCS and the disease signature (Annotation Bar Fig. 2b, and Supplementary Data 4)[13].

**Ranking of single compounds against individual cell expression programs.** The filtered single-cell counts matrix output by CellRanger was log-normalized and scaled using the R package Seurat. Using the normalized consensus signatures from the LINCS L1000 2017 dataset, predicted relative sensitivity of each single-cell to each L1000 perturbation (connectivity) was measured as the Spearman's ρ between a compound's consensus signature and each individual cell's scaled gene expression of the L1000 genes present in each compound's consensus signature (Fig. 2a). Pearson correlation analysis was applied

to the compound versus cell coefficient matrix to visualize small molecule clustering based on integration with the single-cell data (Fig. 2b). Mechanism of action annotations were obtained from the Broad Institute Drug Repurposing Hub (Supplementary Fig. 2)[21]. Single-cell drug connectivities were compared between cell states using a generalized linear model implemented through limma. The mean connectivities of significantly different FDA-approved oncology compounds in addition to alisertib ($p < 0.05$) for each cell state were then scaled and visualized by heatmap (Fig. 2h).

**Tumor cell drug connectivity clustering analysis.** Using the Drug Repurposing Hub, 63 FDA-approved oncology drugs with L1000-derived TCSs from the 2017 data release were identified. A pair-wise Spearman correlation matrix was generated on the calculated drug connectivity values of all tumor cells from each patient in the single-cell atlas for these drugs. Euclidean distance was calculated using the base R function dist(). Ward's hierarchical clustering of the distance matrix was performed using the base R function hclust(). Within cluster variance (sum of squares) was visualized for each number of resulting clusters K using the R package factoextra[102,16]. K of 7 was selected for representative visualizations based on the resulting elbow plot.

**Xenograft tumor cell stratification and state shift analysis.** Xenograft tumor cells were assigned GBM cell state identities using GBM cell transcriptional state signatures (Neftel et al., 2019), as described above for patient data (Fig. 3g, h, and Supplementary Fig. 5b, c). Singscore-derived enrichment scores and proportions of cells predominantly within each transcriptional state were compared between DMSO vehicle and alisertib-treated xenografts (Fig. 3i, j). The analysis pipeline is available at https://github.com/AyadLab/scFOCAL-dataProcessing.

**Patient and xenograft pharmacotranscriptomic comparisons.** To investigate the relationship between cells in GBM22 xenografts and those captured directly from patient tumors, GBM22 cells from DMSO-treated xenografts were input into scFOCAL and were scored for correlation with all small-molecule TCSs. The mean compound-cell connectivity values (Spearman's $\rho$) were calculated across each transcriptional state identity. Using Spearman's $\rho$ once more, these connectivity values were compared to the same mean connectivity values obtained using patient neoplastic cells (Supplementary Fig. 7a). To assess whether scFOCAL predictions of alisertib-resistant and sensitive cell identities from patient data adequately modeled the pharmacotranscriptomic changes occurring in vivo, we utilized a linear model to calculate global L1000 small molecule discordance shift across both patient and xenograft datasets. A pseudo-count of 1 was added to all values in the compound-cell connectivity matrices, which were then fit to a linear model using limma and voom. Patient-based compound-cell connectivity values were grouped by predicted resistance or sensitivity, and differential discordance was calculated across populations within each individual patient tumor. The xenograft cell discordance differential was calculated in the same manner. Resulting logFC values for each compound were compared across in silico and in vivo analyses using Spearman's $\rho$ (Supplementary Fig. 6b) (Supplementary Data 5).

**Panobinostat TCS connectivity analysis in ex vivo GBM acute slice culture.** Single-cell RNA sequencing data of 5 paired acute slice culture samples derived from patient glioblastoma resections treated with either DMSO vehicle or the HDAC inhibitor Panobinostat (0.2 μM) was obtained from GEO accession GSE148842[51]. Single-cell transcriptomes with fewer than 200 detected genes or with greater than the 75th percentile of reads detected, calculated respectively per sample, were kept for downstream analysis. The pass-filter dataset was comprised of

62,250 single-cell transcriptomes with 58,828 features. Individual samples were then integrated using reciprocal PCA within Seurat. SNN clustering and differential expression analysis were used to identify discrete cell types within the integrated dataset. A TCS was generated for panobinostat utilizing L1000 data from within the 2020 release as described above (Supplementary Fig. 7c). Cell connectivity to the panobinostat TCS was calculated across all cells within the dataset. A Wilcoxon rank sum test with continuity correction was utilized to compare panobinostat connectivity across treatment groups within the determined neoplastic (37,682 cells) and myeloid cell (13,017 cells) populations respectively, and was visualized by violin and area-normalized kernel density estimate plots created with ggplot2 (Supplementary Fig. 7d, e). Analysis of proportion change in Neftel et al. states was performed utilizing Q1 and Q3 quantile cut-offs for panobinostat TCS connectivity, and across treatment groups. Comparisons of proportion were performed using a paired Wilcoxon test. Shifts in proportion across these two groupings were compared using the Spearman correlation.

**Combination scoring analysis.** We determined the potential to apply scFOCAL connectivity analysis in ranking synergistic small molecule combinations. A scFOCAL combination score was implemented as the differential logFC of each small molecule's connectivity between cells predicted to be sensitive or resistant to a reference small molecule, multiplied by that small molecule's mean connectivity to cell population predicted to be resistant to the reference small molecule. To assess the utility of this approach, synergy screen data was obtained from Houweling et al. (2023) supplementary data[52]. Raw sum synergy BLISS scores from short-term treatment assays of 24 different cell lines in spheroid culture were used. Following filtering of the synergy screen data for compounds present within the L1000, 4 potential reference small molecules with more than two associated combinations with molecules also in the L1000 data were used for analysis. These reference molecules were erlotinib, lapatinib, pazopanib, and sunitinib. Neoplastic cells from the patient single-cell atlas were scored for connectivity of these and all partner molecules within the screen. Connectivity cut-offs for sensitive or resistant binning of the patient neoplastic cells was calculated as the mean of all cells' connectivity for each reference molecule, respectively. Using limma, the differential connectivity across reference molecule sensitive and resistant cell populations, for each of the partner compounds present in the screen, were calculated for each individual patient within the atlas. The resulting logFC values from each patient for each partner molecule were averaged across all individual patients. These aggregate differential connectivities were then multiplied by the mean connectivity score of each respective small molecule to the resistant cell population to the respective reference small molecule across all patients. These scores were then scaled between 0 and 1. Calculated combination score and observed BLISS raw sum synergy for each combination was compared using Spearman correlation in each of the 24 cell lines tested, and across the aggregate mean BLISS scores for all cell lines for each combination.

**Scoring of L1000 small molecules for predicted synergy with CT-179.** CT-179 single-cell connectivity was calculated as the Spearman correlation between the normalized and scaled expression cells within our dataset against the $\log_2$FC calculated for 200 nM CT-179 treatment of GBM8 cells. As above, sensitive and resistant cells were defined as those having CT179 connectivities less than and greater than 0 respectively. L1000 drug-cell connectivities were transformed using the Fisher's Z transformation, and limma was used to calculate the differential connectivities to L1000 molecules between predicted CT-179 sensitive and resistant cells, using patient ID as a latent variable. Compounds with logFC <0.5 and FDR < 0.05 were retained. The logFC values for these compounds were then weighted by multiplying by

their respective mean resistant cell-drug connectivities to derive our scFOCAL-based combination index. Thus, a more positive combination index value indicates both decreased connectivity to that drug in the CT-179 reference population and an overall discordance of the drug with CT-179 predicted resistant cells. This combination index can be defined as follows:

$$Combination\ Index = \log_2 FC \times \bar{\rho} \times I(\log_2 FC < 0\ and\ \bar{\rho} < 0) \quad (1)$$

Where a combination index is only calculated when both the differential connectivity ($\log_2 FC$) and the mean resistant cell connectivity (RCC) ($\rho$) are negative values. This combination index is a feature implemented within the scFOCAL R package and associated shiny GUI.

### Additional resources
scFOCAL Shiny Web GUI: https://robert-k-suter.shinyapps.io/scFOCAL.
scFOCAL R package: https://github.com/AyadLab/scFOCAL

### Reporting summary
Further information on research design is available in the Nature Portfolio Reporting Summary linked to this article.

### Data availability
All data needed to generate the conclusions in this manuscript are present in the paper and/or the Supplementary Information. Our generated single-cell data of patient GBM tumors, including processed Seurat objects, are available via the Gene Expression Omnibus (GEO) with accession ID GSE229779. Xenograft single-cell data, including processed Seurat objects, are also available via the Gene Expression Omnibus (GEO) with accession ID GSE231489. Raw and processed bulk RNA-sequencing data from CT-179 and vehicle-treated GBM8 cells are available via GEO with accession ID GSE300210. These GEO series are under GEO SuperSeries GSE231490. Additional patient single-cell data from Johnson et al. was obtained from https://synapse.org/singlecellglioma, and acute slice culture single-cell data was obtained from GEO at GSE148842. The remaining data are available within the Article, Supplementary Information or Source Data file. Source data are provided with this paper.

### Code availability
Data, code, and materials are available to any researcher, and processing pipelines can be found at https://github.com/AyadLab/scFOCAL-dataProcessing or within the scFOCAL package source code available at https://github.com/AyadLab/scFOCAL, which contains a comprehensive readme file, tutorials, and example datasets for use with the R package-based application. A permanent, citable version of the code is available via Zenodo at https://doi.org/10.5281/zenodo.17555928.100.

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

## Acknowledgements

The authors thankfully acknowledge Dr. Corneliu Sologon, Dr. Jenny Kemper, Marissa Brooks and Yoslayma Cardentey of the Sylvester Comprehensive Cancer Center Oncogenomics Shared Resource (OGSR). We thank the Georgetown University Lombardi Cancer Center, the Georgetown University Department of Veterinary Resources, the Georgetown University Information Services, the University of Miami Department of Veterinary Resources, and the Miami Project to Cure Paralysis. We thank the Mayo Clinic Brain Tumor Patient-Derived Xenograft National Resource. We thank Luke Tallon and Lisa Sadjewicz of the University of Maryland Institute for Genome Sciences. We thank Dr. Vance Lemmon, Dr. John Bixby, Dr. Claes Wahlestedt, Dr. William J. Harbour, Dr. Daniel Pelaez, Dr. David J. Robbins, and all members of the Schürer and Ayad laboratories for invaluable discussions of these studies. Additionally, we would like to acknowledge the following funding sources:

Funding from the Sylvester Comprehensive Cancer Center, FCBTR, and FACCA to NGA

Funding from the Sylvester Comprehensive Cancer Center to SCS
Funding from the Sylvester Comprehensive Cancer Center to MID
Funding from the Lombardi Comprehensive Cancer Center to RKS
Funding from the Lombardi Comprehensive Cancer Center to NGA
Funding from the Lombardi Comprehensive Cancer Center Cancer Cell Biology (CCB) Program to RKS and NGA

Funding from NIH RM1NS133003 (NINDS) to NGA and RKS
Funding from NIH P30CA051008 (NCI)
Funding from NIH P30CA240139 (NCI), U54HL127624 (LINCS program, through NHLBI).

Funding from Bellringer

American Cancer Society Institutional Research Grant (IRG-23-1156148-27-IRG, pilot award to RKS, PI: Riggins).

This work used Jetstream2 at Indiana University through allocation BIO240140 from the Advanced Cyberinfrastructure Coordination Ecosystem: Services & Support (ACCESS) program, which is supported by National Science Foundation grants #2138259, #2138286, #2138307, #2137603, and #2138296.

## Author contributions

Conceptualization: R.K.S., A.M.J., V.S., and N.G.A.; Methodology, investigation, visualization: R.K.S.; Tissue Acquisition: M.E.I, R.J.K.; scRNAseq: R.K.S., S.L.W; In vitro studies: A.W., S.K.; Computational design: R.K.S.; Data analysis: R.K.S., R.V., S.K., M.D., M.S., L.Z., R.C., P.P.; App development: R.K.S., A.M.J., R.V., M.D., R.C., G.B., L.R.; Animal studies: R.K.S., A.M.J., S.H.H., S.K., W.W., M.C., D.B., E.B.R., A.O., M.G.A., J.N.S; Writing and revision: R.K.S., A.M.J., S.K., R.V., M.D., M.I.D, M.E.I., R.J.K., G.S., S.K., N.G.A.; Supervision: S.C.S., N.G.A.

## Competing interests

The authors declare no competing interests.

## Additional information

[1]Department of Oncology, Lombardi Comprehensive Cancer Center, Georgetown University, Washington, DC, USA. [2]Department of Molecular and Cellular Pharmacology, Institute for Data Science & Computing, University of Miami Miller School of Medicine, Miami, FL, USA. [3]Sylvester Comprehensive Cancer Center, University of Miami Miller School of Medicine, Miami, FL, USA. [4]Department of Neurological Surgery, University of Miami Miller School of Medicine, Miami, FL, USA. [5]Department of Neurology, University of Miami Miller School of Medicine, Miami, FL, USA. [6]Department of Radiation Oncology, Mayo Clinic, Rochester, MN, USA. [7]AbbVie, Oncology Discovery, North Chicago, IL, USA. [8]Curtana Pharmaceuticals, Inc., Austin, TX, USA. [9]Pacific Neuroscience Institute and Saint John's Cancer Institute, Providence Health System, Santa Monica, CA, USA. [10]Asthra Health and Lundquist Institute, Santa Monica, CA, USA. ✉e-mail: RKS82@Georgetown.edu; NA853@Georgetown.edu

