## [Transparent Peer Review file · Nature Communications]

Drug and single-cell gene expression integration identifies sensitive and resistant glioblastoma cell populations

Corresponding Author: Dr Nagi Ayad

Version 0:

Reviewer comments:

Reviewer #1

(Remarks to the Author)

In this manuscript, Suter et al. developed a framework to identify cell type specific effective drugs for glioblastoma (GBM) using patient derived scRNAseq and drug induced transcriptional profiles from LINCS L1000 dataset. Their method, named ISOSCELES, uses cell type specific single cell transcriptomics data, and is based on the “drug induced signature reversal” principle. Using signature reversal methods with single cell data is an important step for the applicability of these methods, so the manuscript can be interesting for the systems biology community. The authors also provide an R package and a Shiny app for ISOSCELES. I have mainly methodological comments, and also some regarding the released tool.

Major:

In a recent paper (Koudijs et al., Validation of transcriptome signature reversion for drug repurposing in oncology, Briefings in Bioinformatics 2023) a potential issue regarding drug repurposing approaches in oncology has been raised. While tumor cells have a proliferation signature, oncology drugs have an anti-proliferation signature, so the observed anti-similarity can be aspecific, only a consequence of trivial difference between proliferative and anti-proliferative signature. The authors should discuss the potential effect of this fact on their results.

More specifically, I assume that alisertib strongly inhibits cell cycle. Based on previous studies (e.g.: the already cited Nefel et al. Cell 2019 paper), NPC-like cells have the largest proportion of cycling cells, while MES-like cells have the lowest proportion. Couldn't explain the observed changes in cell proportions after alisertib changes simply the fact that alisertib has anti-similarity to cycling cell signature (more abundant in NPC-like cells), and inhibits this cycling cell population, thus decreasing their proportion? A more detailed analysis of cell cycle score in different populations could have help to discard this possibility.

In the “In silico drug connectivity analysis predicts intratumor compound discordance shifts seen in vivo” section it is hard to understand what is the consequence of these results. Could the authors define Observed / Predicted Corr. Coefficient FC more precisely (SFigure 5)? If I understand correctly, here the authors compare predicted cell proportion changes, for a given compound, and observed proportion changes after alisertib treatment? What is the reasoning behind this comparison? Also, the drugs with high correlation in SFigure 5 are strong inhibitors of cell cycle (tubulin polymerization / topoisomerase inhibitors), which can be connected to the two points above.

Minor:

The GitHub repository for the manuscript is currently empty, I would suggest sharing it with the reviewers during the review process.

For ISOSCELES Shiny application some tutorial dataset would be helpful.

Reviewer #2

(Remarks to the Author)

The manuscript describes application of disease signature reversal to the transcriptional profiles of single cells as a prediction of drug sensitive and resistant populations in glioblastoma tumors. Based on scRNAseq from 11 GBM cases (a

mix of primary and recurrent cases), the authors derive disease signatures for each of four GBM transcriptional cell states. The reversal of these disease signatures in L1000 small molecule transcriptomic dataset is then calculated to predict which subpopulation of GBM cells could be more sensitive or resistant to specific drugs. Conceptually, this is a very interesting approach, which could help identify drug combinations or sequence in which drugs should be combined to more effectively target all subpopulations in GBM. However, the current manuscript provides only a single example of a drug for which the computational analysis was performed and only a single PDX for experimental validation. It is thus unclear how robust is the ISOSCELES platform.

Specific comments:

1. In the example provided, it is mentioned that several drugs are predicted to target the Alisertib resistant population. It would be important to demonstrate if these predictions are correct by testing these drugs on resistant populations in vitro or combination treatment in vivo.
2. The ISOSCELES figure (Fig. 5, Results, Proportion shift tab) suggests that there are two population expanding as resistant – MES1 and AC. How can this be reconciled with results presented in Fig. 4? Experiments performed on another cell line or two could help determine whether these expansions depend on initial fraction of the cell state in untreated tumor.
3. The current ISOSCELES model is built on 11 datasets, of which 4 are recurrent GBM cases. Would the predictions be better if only primary cases were used? Would they look different for only recurrent cases? There is several publicly available datasets based on 10xGenomics platform that could be used to test this: ex. 16 patients in PMID: 32641768 , 10 patients in PMID: 35122077, 11 patients in PMID: 31901251 and many others.
4. The overall robustness of the study would be significantly strengthened by additional PDXs tested for their Alisertib response. In a heterogeneous disease, such as GBM, a single cell line/PDX might not be representative.

Minor comments:

1. Please include a summary of the number of cells and gene count per cell to describe the single-cell datasets used in this study.
2. CytoTRACE-based assignment of malignant and non-malignant state should be briefly explained in the text to clarify what is presented in Fig. 1d.
3. Reference to Fig.3a in the text is missing.
4. Fig. 3 – would supervised clustering based on disease signature reversal specific to each cell state show significant differences between states? In other words, is L1000 dataset integration with expression data helpful as a discovery tool?
5. Fig. 4g – is the change in percent composition significantly different after Alisertib treatment? Please provide statistical test p value.

Reviewer #3

(Remarks to the Author)

In this manuscript, the authors present an interesting general approach: the development of an algorithm that allows to predict sensitivity of glioblastoma cell subpopulations (i.e., NPC-, OPC-, MES- and AC-like cells) to certain drugs and drug classes. Tumor heterogeneity is a formidable challenge in the treatment of glioblastoma, and a better idea of which transcriptional tumor cell subpopulations respond to which drugs/classes can help to develop tailored multi- or oligo-drug strategies; potentially even identifying one drug that is able to target all, or at least the most important, tumor cell subpopulations.

Having said that, this manuscript falls short of providing evidence that the computational in silico approach presented here is of any practical (biological / therapeutic / mechanistic) value. The investigation of one single drug in one single cell line in vivo demonstrating one major change - a shift towards MES-like tumor cell subpopulation - is no evidence for the utility of this entire approach. A shift towards MES-like states is well known for many therapeutic interventions now, and far away from any proof of a specific alteration induced.

The authors would need to demonstrate that their predictions regarding multiple drug classes is really true, at least relevantly overlapping with experimental data. They need to study the response of all 4 subpopulations to a panel of drugs experimentally; subpopulations can be studied by enrichment/depletion analyses, better individually after FACS sorting or, ideally, and something this reviewer would strongly recommend, using in vivo reporter systems or bar code systems that allow to distinguish all 4 subpopulations. This should be done in state-of-the-art in vitro assays. After that, the authors need to demonstrate in vitro and in vivo that the most promising compound (mix) is indeed providing a therapeutic advantage compared to other approaches. The results need to be clear and convincing to make this study to a relevant advance in the field. The authors need to provide evidence that ISOSCELES is of any de facto meaning and usability for the community. This most important part is simply missing from this study.

More minor point: the presentation of the results and figure legends is often confusing and should be significantly improved. All figure panels should be cited in the order of appearance; it should be explained in the results section what they actually show; and the reader needs to get more explanatory informations there, too.

Version 2:

Reviewer comments:

Reviewer #1

(Remarks to the Author)

Thank the authors for answering my questions, and also for the rich additional analysis. I think the synergy prediction strongly increased the importance of this paper.

(Remarks on code availability)

Reviewer #2

(Remarks to the Author)

This is a revised manuscript describing a new computational tool for predicting drug sensitivity at the single cell level. The main point raised in the prior round of review was the robustness of the prediction, as limited experimental data were shown, and the question about the ability to predict combinations of drugs to target the subpopulations.

The presented approach to identify drug combinations is highly warranted. However, the scFOCAL wouldn't have been needed to come up with a combination of OLIG2i and EGFR-MMAF, since both targets are what distinguishes the transcriptional cell states in GBM. Docetaxel and varlitinb don't seem to be the top hits in the connectivity analysis. It would be important for the readers to understand how to prioritize hits that may be selected when they apply scFOCAL to their own dataset.

Since this manuscript has been significantly altered from the previous version, there are also several other points that should be addressed:

1. CytoTRACE should be better described in the text, as it is unclear what it represents in Fig. 2f&g.
2. Line 168: "Within each individual, cycling cells seem to separate into a unique drug connectivity state". Is this true for all patients in this cohort or only for the two tumors presented in Fig. 2?
3. The authors focus on alisertib, based on previous connection to MES cell state. There are a number of other candidates, including alpelisib and afatinib, which have highly differential connectivity between cell states. This should be clarified.
4. Fig.3a. Predicted sensitive vs resistant populations are divided right at 0 point for TCS correlation coefficient. Would the differences be more striking if top and bottom quartiles were used? With the current split, in Fig.3b the distribution looks very similar.
5. Fig. 3j. Is the increase in MES and decrease in NPC after alpelisib treatment statistically significant? The changes are not massive, so statistical test would be helpful.
6. Fig. 3k. The correlation between observed and predicted shifts is weak overall, yet for some classes of compounds they work well (Fig. S6). It might be worth to put S6 panels in the main figure, since they represent a more meaningful result. Is there any connection between these drug classes and action or transcriptional consequences of alpelisib treatment?
7. The Panobinostat response analysis seems a bit disconnected from the rest of the manuscript. Since this is an scRNAseq dataset, are the authors able to see changes in Neftel et al. cell states in the cultured slices?
8. Docetaxel and varlitinb don't seem to come up as top hits in the connectivity analysis. Could the authors comment on the top hits and why they were not pursued for experimental validation?
9. Fig. 2b – scale for disease signature reversal is missing

(Remarks on code availability)

Reviewer #3

(Remarks to the Author)

The authors have responded well to many of my comments and have now provided a better proof-of-principle that their novel cell state - based approach can indeed help to identify new therapeutic combinations in GBM.

(Remarks on code availability)

Version 3:

Reviewer comments:

Reviewer #2

(Remarks to the Author)

The authors clarified all of the points raised in review. The manuscript will be of high interest to Nature Communications readers hoping to streamline synergistic drug combination testing.

(Remarks on code availability)

Reviewer 1: Computational pharmacology, omics (Remarks to the Author): In this manuscript, Suter et al. developed a framework to identify cell type specific effective drugs for glioblastoma (GBM) using patient derived scRNAseq and drug induced transcriptional profiles from LINCS L1000 dataset. Their method, named ISOSCELES, uses cell type specific single cell transcriptomics data, and is based on the “drug induced signature reversal” principle. Using signature reversal methods with single cell data is an important step for the applicability of these methods, so the manuscript can be interesting for the systems biology community. The authors also provide an R package and a Shiny app for ISOSCELES. I have mainly methodological comments, and also some regarding the released tool.

In a recent paper (Koudijs et al., Validation of transcriptome signature reversion for drug repurposing in oncology, Briefings in Bioinformatics 2023) a potential issue regarding drug repurposing approaches in oncology has been raised. While tumor cells have a proliferation signature, oncology drugs have an anti-proliferation signature, so the observed anti-similarity can be aspecific, only a consequence of trivial difference between proliferative and anti-proliferative signature. The authors should discuss the potential effect of this fact on their results. More specifically, I assume that alisertib strongly inhibits cell cycle. Based on previous studies (e.g.: the already cited Nefel et al. Cell 2019 paper), NPC-like cells have the largest proportion of cycling cells, while MES-like cells have the lowest proportion. Couldn't explain the observed changes in cell proportions after alisertib changes simply the fact that alisertib has anti-similarity to cycling cell signature (more abundant in NPC-like cells), and inhibits this cycling cell population, thus decreasing their proportion? A more detailed analysis of cell cycle score in different populations could have help to discard this possibility.

We thank the reviewer for their important perspective on the consideration of the cell cycle in the use of disease signature reversal approaches. In our analysis, enrichment scores for Neftel states in both the presented patient GBM scRNAseq atlas and within *in vivo* studies were calculated using cell cycle expression regressed scaled data. Regression was performed within the ScaleData() function of Seurat on G2M and S phase module scores as calculated as in Tirosh et al., 2015, as implemented within the CellCycleScoring() function of Seurat, using updated G2M and S phase gene sets as provided within the Seurat package. This function also bins cells into their cell cycle phase identities based on the G2/M and S phase module scores, and the proportions of cells in each phase within predicted alisertib sensitive or resistant cell populations is depicted in Supplementary Figure S3d. In both the predicted resistant population from the patient GBM scRNAseq atlas, and in the alisertib treated xenografts, the proportion of cells in both G2/M and S are increased relative to the predicted alisertib sensitive population or the DMSO treated xenografts respectively; suggestive that the predicted and observed shift from NPC- to more MES-like upon alisertib treatment is not driven by a decrease in proliferative cells. Importantly, alisertib has been reported to arrest cells in S and G2/M phases.

In the “In silico drug connectivity analysis predicts intratumor compound discordance shifts seen in vivo” section it is hard to understand what is the consequence of these results. Could the authors define Observed / Predicted Corr. Coefficient FC more precisely (SFigure 5)? If I understand correctly, here the authors compare predicted cell proportion changes, for a given compound, and observed proportion changes after alisertib treatment? What is the reasoning behind this comparison? Also, the drugs with high correlation in SFigure 5 are strong inhibitors of cell cycle (tubulin polymerization / topoisomerase inhibitors), which can be connected to the two points above.

We thank the reviewer for their important comments on the presentation of the drug connectivity analysis. For Supplementary Figure S5, we sought to characterize the pharmacotranscriptomic differences between cells predicted to be sensitive or resistant to alisertib treatment using ISOSCELES. Here, we identified small molecules with a differential drug-cell connectivity (cell expression vs drug TCS) between predicted alisertib sensitive (Alisertib TCS Connectivity < 0) and resistant (Alisertib TCS Connectivity > 0) cell populations using limma, within each individual patient in the dataset. Calculated results were aggregated from each individual patient to a consensus of which small molecules had conserved shifts across all patient predicted sensitive and resistant cell populations. We then repeated this analysis using the data from our in vivo experiments to identify small molecules with differential drug-cell connectivity scores across PDX cell populations divided by treatment arm.

The thought behind this analysis is now more complete and leads to subsequent predictions of potential synergistic combinations based on a reference small molecule, where the reference molecule connectivity divides single-cell input into resistant and sensitive populations, used in the calculation of an ISOSCELES based combination score discussed in more detail further down in this rebuttal.

Minor: The GitHub repository for the manuscript is currently empty, I would suggest sharing it with the reviewers during the review process. For ISOSCELES Shiny application some tutorial dataset would be helpful.

We thank the reviewer for pointing this out. The github repository for the ISOSCELES application itself contains the ISOSCELES R package source code, and is available at <https://github.com/AyadLab/ISOSCELES>. Within the readme file for this git, are links to download tutorial datasets for use with the application. We have also since deposited all processing pipelines, including those for analyses performed during revision, within <https://github.com/AyadLab/ISOSCELES-dataProcessing>. While the application is available on shinyapps.io, we suggest installing as an R package as per the readme file in the github repository, as analysis with the tutorial dataset will run more efficiently.

Reviewer 2: The manuscript describes application of disease signature reversal to the transcriptional profiles of single cells as a prediction of drug sensitive and resistant populations in glioblastoma tumors. Based on scRNAseq from 11 GBM cases (a mix of primary and recurrent cases), the authors derive disease signatures for each of four GBM

transcriptional cell states. The reversal of these disease signatures in L1000 small molecule transcriptomic dataset is then calculated to predict which subpopulation of GBM cells could be more sensitive or resistant to specific drugs. Conceptually, this is a very interesting approach, which could help identify drug combinations or sequence in which drugs should be combined to more effectively target all subpopulations in GBM. However, the current manuscript provides only a single example of a drug for which the computational analysis was performed and only a single PDX for experimental validation. It is thus unclear how robust is the ISOSCELES platform.

Specific comments: Minor comments:

Fig. 3 – would supervised clustering based on disease signature reversal specific to each cell state show significant differences between states? In other words, is L1000 dataset integration with expression data helpful as a discovery tool?

Thank you for this comment. We agree that L1000 TCS and scRNA-seq integration may be useful as a discovery tool, however we were not able to address this within the scope of this manuscript.

Fig. 4g – is the change in percent composition significantly different after Alisertib treatment? Please provide statistical test p value.

We thank the reviewer for this important comment. As individual xenograft samples with shared treatment were pooled together we are unable to perform a statistical significance assessment in regards to the percent composition before and after treatment. However, at the single-cell level, we are able to claim that the overall enrichment for the MES-like module is statistically greater, and that NPC-like expression is significantly depleted in alisertib treated xenografts. This has been added to Figure 5f.

In the example provided, it is mentioned that several drugs are predicted to target the Alisertib resistant population. It would be important to demonstrate if these predictions are correct by testing these drugs on resistant populations in vitro or combination treatment in vivo.

We thank the reviewer for this important perspective on the utility of the ISOSCELES platform to identify combinations of small molecules which are predicted by the platform to target unique cell populations present within the same GBM tumor. To this end, during revisions we have refined the algorithm within the ISOSCELES platform for identifying potentially synergistic combinations and visualization of its output. We applied this algorithm to the patient scGBM atlas used prior to generate combination scores for compounds used within the high-throughput synergy screen published in Houweling et al, 2023. This synergy screen was performed on 25 different glioma stem cell lines in spheroid culture. Raw sum synergy measured by BLISS was compared to combination scores for tested molecules in combination with either lapatinib or pazopanib. In both cases, overall, combination score was moderately correlated with observed BLISS raw sum synergy. For the majority of cell lines tested, this combination score was very

predictive of relative synergy of the combinations tested. The most potent combinations in these instances were of those made of compounds with distinct TCS connected cell populations.

The ISOSCELES figure (Fig. 5, Results, Proportion shift tab) suggests that there are two population expanding as resistant – MES1 and AC. How can this be reconciled with results presented in Fig. 4? Experiments performed on another cell line or two could help determine whether these expansions depend on initial fraction of the cell state in untreated tumor.

Thank you for bringing this to our attention. This figure was meant to be illustrative and was based on an outdated testing dataset. We have updated Figure 6 to illustrate ISOSCELES analysis of the sample data included with the GEO release to resolve the confusion.

The current ISOSCELES model is built on 11 datasets, of which 4 are recurrent GBM cases. Would the predictions be better if only primary cases were used? Would they look different for only recurrent cases? There is several publicly available datasets based on 10xGenomics platform that could be used to test this: ex. 16 patients in PMID: 32641768 , 10 patients in PMID: 35122077, 11 patients in PMID: 31901251 and many others.

Thank you for your comment. We sought to develop predictions regardless of newly diagnosed or recurrent tumor status, as many patients seek out trials upon tumor recurrence. As such, we have not tried to separate newly diagnosed from recurrent tumors since we do not have enough recurrent GBM patient tumors for these predictions.

The overall robustness of the study would be significantly strengthened by additional PDXs tested for their Alisertib response. In a heterogeneous disease, such as GBM, a single cell line/PDX might not be representative.

In order to highlight the utility of the study, we have applied ISOSCELES to scRNAseq data obtained from GSE of acute slice culture derived from patient GBMs treated with either DMSO or the HDAC inhibitor panobinostat, and showed that ISOSCELES calculated panobinostat connectivity of single-cells correlates with panobinostat treatment *ex vivo*.

Please include a summary of the number of cells and gene count per cell to describe the single-cell datasets used in this study.

Thank you for addressing this important oversight. We have since added cell and detected gene counts within the manuscript for patient, xenograft, and newly analyzed acute slice culture scRNAseq datasets.

CytoTRACE-based assignment of malignant and non-malignant state should be briefly explained in the text to clarify what is presented in Fig. 1d.

Thank you for your comment. We have updated the methods with a brief explanation of CytoTRACE to clarify the importance of Fig. 1d.

Reference to Fig.3a in the text is missing.

Thank you for bringing this to our attention. Figure 3a has now been referenced in text.

Reviewer 3: GBM therapy, preclinical models (Remarks to the Author): In this manuscript, the authors present an interesting general approach: the development of an algorithm that allows to predict sensitivity of glioblastoma cell subpopulations (i.e., NPC-, OPC-, MES- and AC-like cells) to certain drugs and drug classes. Tumor heterogeneity is a formidable challenge in the treatment of glioblastoma, and a better idea of which transcriptional tumor cell subpopulations respond to which drugs/classes can help to develop tailored multi- or oligo-drug strategies; potentially even identifying one drug that is able to target all, or at least the most important, tumor cell subpopulations. Having said that, this manuscript falls short of providing evidence that the computational in silico approach presented here is of any practical (biological / therapeutic / mechanistic) value.

The investigation of one single drug in one single cell line in vivo demonstrating one major change - a shift towards MES-like tumor cell subpopulation - is no evidence for the utility of this entire approach. A shift towards MES-like states is well known for many therapeutic interventions now, and far away from any proof of a specific alteration induced.

We thank the reviewer for their important comment. We agree that as this tool was developed, the understanding of a common proneural-to-mesenchymal transcriptional response in glioma has been increasingly characterized. In line with comments from another reviewer, we have addressed this issue by identifying different small molecules which are predicted to have varying responses to that predicted and observed in response to alisertib. Importantly, not all proneural to mesenchymal programs are the same, and differing NPC-like and MES-like states have already been characterized. While not within the scope of this paper, more mechanistic studies on the proneural to mesenchymal response in GBM need to be performed.

To highlight the utility of ISOSCELES in predicting heterogeneous cell population response outside of a proneural to mesenchymal shift, we have utilized ISOSCELES to predict dynamics of the tumor associated macrophage population within glioma slice culture treated with the HDAC inhibitor panobinostat. Panobinostat TCS connectivity very effectively separates DMSO and panobinostat treated macrophages on a probabilistic distribution.

They need to study the response of all 4 subpopulations to a panel of drugs experimentally; subpopulations can be studied by enrichment/depletion analyses, better individually after FACS sorting or, ideally, and something this reviewer would strongly recommend, using in vivo reporter systems or bar code systems that allow to distinguish all 4 subpopulations. This should be done in state-of-the-art in vitro assays.

We thank the reviewer for these suggestions. Given that the editor asked for the revised manuscript within three months we focused on analyzing existing datasets, which supported our

novel platform ISOSCELES. We were unable to perform more in vivo experiments within the 3-month period using other drugs.

After that, the authors need to demonstrate in vitro and in vivo that the most promising compound (mix) is indeed providing a therapeutic advantage compared to other approaches.

We thank the reviewer for this suggestion. Given that the editor asked for the manuscript back within 3 months it was not possible to perform more in vivo experiments. However, we have mined previously published data for panobinostat in slice cultures and we demonstrate that the ISOSCELES predictions are consistent with what was observed by single cell analysis ex vivo. These studies further support that our pipeline predicts the cells that are targeted pharmacologically in GBM. We have also extended this finding further using mined synergy screen data from Houwelling et al, 2023. Here, we score small molecule combinations using ISOSCELES, and found that our predictions correlate with relative synergy results across 24 GBM cell lines in spheroid culture.

More minor point: the presentation of the results and figure legends is often confusing and should be significantly improved. All figure panels should be cited in the order of appearance; it should be explained in the results section what they actually show; and the reader needs to get more explanatory informations there, too.

We thank the reviewer for these suggestions and have now modified the order of the figures to be consistent with the order of appearance. We have also explained in depth what the results show as suggested in the figure legends and results.

Response to Reviewers:

Reviewer 1: Computational pharmacology, omics (Remarks to the Author): In this manuscript, Suter et al. developed a framework to identify cell type specific effective drugs for glioblastoma (GBM) using patient derived scRNAseq and drug induced transcriptional profiles from LINCS L1000 dataset. Their method, named ISOSCELES, uses cell type specific single cell transcriptomics data, and is based on the “drug induced signature reversal” principle. Using signature reversal methods with single cell data is an important step for the applicability of these methods, so the manuscript can be interesting for the systems biology community. The authors also provide an R package and a Shiny app for ISOSCELES. I have mainly methodological comments, and also some regarding the released tool.

In a recent paper (Koudijs et al., Validation of transcriptome signature reversion for drug repurposing in oncology, Briefings in Bioinformatics 2023) a potential issue regarding drug repurposing approaches in oncology has been raised. While tumor cells have a proliferation signature, oncology drugs have an anti-proliferation signature, so the observed anti-similarity can be aspecific, only a consequence of trivial difference between proliferative and anti-proliferative signature. The authors should discuss the potential effect of this fact on their results. More specifically, I assume that alisertib strongly inhibits cell cycle. Based on previous studies (e.g.: the already cited Nefel et al. Cell 2019 paper), NPC-like cells have the largest proportion of cycling cells, while MES-like cells have the lowest proportion. Couldn't explain the observed changes in cell proportions after alisertib changes simply the fact that alisertib has anti-similarity to cycling cell signature (more abundant in NPC-like cells), and inhibits this cycling cell population, thus decreasing their proportion? A more detailed analysis of cell cycle score in different populations could have help to discard this possibility.

We thank the reviewer for their important perspective on the consideration of the cell cycle in the use of disease signature reversal approaches. We have made our best effort to take cell-cycle-related gene expression differences into consideration throughout our different analyses. First, enrichment scores for Neftel states in both the presented patient GBM scRNAseq atlas and *in vivo* studies were calculated using cell cycle-regressed and scaled data. Regression was performed within the ScaleData() function of Seurat on G2M and S phase module scores calculated as in Tirosh et al., 2015, as implemented within the CellCycleScoring() function of Seurat, using updated G2M and S phase gene sets as provided within the Seurat package. This function also bins cells into their cell cycle phase identities based on the G2/M and S phase module scores, and the proportions of cells in each phase within predicted alisertib-sensitive or resistant cell populations are depicted in Supplementary Figure S4d. Further, as described in the methods section, scFOCAL cell-drug connectivity is calculated against cell-cycle-regressed single-cell expression.

Supplementary Figure S4d: Sankey diagrams depicting proportions of predicted sensitive and resistant cell populations across cell cycle phase.

Comparing the cell-cycle differences between the predicted sensitive and resistant cell populations, it can be seen that an abundance of cells determined to be in G1 are predicted to be alisertib-sensitive, also that an abundance of cells determined to be in S-phase are predicted to be resistant. Importantly, alisertib has been reported to arrest cells in S and G2/M phases. In both the predicted resistant population from the patient GBM scRNAseq atlas and the alisertib-treated xenografts, the proportions of cells in both G2/M and S are increased relative to the predicted alisertib-sensitive population or the DMSO-treated xenografts, respectively. Taken together, our prediction and result *in vivo* that NPC-like cells are preferentially depleted by alisertib does not seem to be an artifact related to cell-cycle-related gene expression or a decrease in proliferative cells.

Further, to demonstrate more effectively that our platform is identifying biology beyond the cell cycle, we now include a second *in vivo* experiment to demonstrate that our approach is useful for predicting drugs targeting different GBM cell transcriptional states in tumors. We included a new *in vivo* study with the Olig2 inhibitor CT-179, as we expected this compound to preferentially target the OPC-like GBM cell state. Our new bulk RNA sequencing data also indicates an increase in proliferation and cell-cycle-related pathways in GBM8 Mayo PDX cells after treatment with CT-179 (**Supplementary Figure S9a-b**).

Supplementary Figure S9: Generation of a GBM CT-179 transcriptional response signature.

a. Volcano plot of differential expression between 200nM CT-179 and vehicle-treated GBM8 cells calculated via DESeq2. **b.** Barplot depicting gene ontology analysis of the filtered CT-179 transcriptional response signature. Bars are colored by the respective ontology used. Selected GO terms highlighted include ‘microtubule’, ‘microtubule binding’, ‘tubulin binding’, and ‘microtubule motor activity’.

Via scFOCAL, as expected, CT-179 is predicted to target OPC-like cells. Interestingly, though, CT-179 is predicted to yield an AC-like resistant cell population. Through the prioritization of EGFR and tubulin targeting molecules by our combination prioritization strategy, which we previously included in our original submission as Supplementary Figure S6, we validate our predictions from a CT-179 response signature by demonstrating synergy *in vivo* with CT-179 and Depatux-M, an anti-EGFR MMAF (tubulin inhibitor) antibody-drug conjugate (**Fig. 4, Fig. 5, Fig. S9**). While further mechanistic investigation of the mechanism behind this synergy is required, we hypothesize that Depatux-M is targeting a more proliferative AC-like population of cells that express EGFR. We sincerely hope that this new use-case demonstration and *in vivo* validation of our scFOCAL platform’s performance ease caution on cell-cycle related bias in its application.

Importantly, our alisertib experiment still fits the altered narrative of this manuscript through demonstration that *in silico* perturbation of our patient GBM scRNAseq atlas accurately predicts differential connectivities to other drugs between the vehicle and alisertib-treated PDX tumors (**Fig. 3k, below**), which led us to investigate the potential for our framework to prioritize synergistic combinations. (Also see added *in vitro* validation of our scFOCAL combination score performed using data obtained from Houweling et al, 2023 (Neuro Oncology Advances).)

Figure 3k. Scatterplot of L1000 small molecule TCS's depicting predicted correlation shift (\log_2FC) vs. observed correlation shift (\log_2FC) in alisertib-treated xenografts. Differential small molecule correlations were calculated using *limma*.

In the “In silico drug connectivity analysis predicts intratumor compound discordance shifts seen in vivo” section it is hard to understand what is the consequence of these results. Could the authors define Observed / Predicted Corr. Coefficient FC more precisely (SFigure 5)? If I understand correctly, here the authors compare predicted cell proportion changes, for a given compound, and observed proportion changes after alisertib treatment? What is the reasoning behind this comparison? Also, the drugs with high correlation in SFigure 5 are strong inhibitors of cell cycle (tubulin polymerization / topoisomerase inhibitors), which can be connected to the two points above.

We thank the reviewer for their important comments on the presentation of the drug connectivity analysis. For the original Supplementary Figure S5, we sought to characterize the pharmacotranscriptomic differences between the cells predicted to be sensitive or resistant to alisertib treatment using scFOCAL. Importantly, these predicted sensitive and resistant populations should be considered analogous to the DMSO-treated or alisertib-treated cells, respectively, from tumors in our *in vivo* experiment.

In our revisions, we have made this point a focus, as herein lies the potential to utilize differential drug connectivity to identify more effective drug combinations. As such, we have moved this figure from the supplement to a main figure (**Fig. 3k**). Here, we identified small molecules with a differential drug-cell connectivity (cell expression vs drug TCS) between predicted alisertib-sensitive (Alisertib TCS Connectivity < 0) and -resistant (Alisertib TCS Connectivity > 0) cell populations using *limma*, within each **patient sample** in the dataset. These differentials represent our predictions of alisertib-induced increases or decreases in sensitivities to other L1000 molecules. Calculated results were aggregated from each patient to a consensus of which small molecules had conserved shifts across all patients' predicted sensitive and resistant cell populations. We then repeated this analysis using the data from our *in vivo* alisertib experiments to identify small molecules with differential drug-cell connectivity scores across actual treatments (DMSO vs alisertib). In our new Fig 3k (previously supplemental), we demonstrate a significant correlation across all L1000 molecules of their predicted connectivity

differentials determined *in silico* vs those differentials calculated from actual treatment data. **This analysis serves to show that the cells which we are predicting to be sensitive or resistant from our patient dataset are transcriptionally (in the context of L1000 signatures) similar to the tumor cells that are sensitive or resistant (persisting after treatment) respectively in our xenograft experiments, and serves as a basis for our new scFOCAL combination index algorithm.** Importantly, this differential drug connectivity measure is utilized within our combination scoring approach, as depicted below in **Fig. 4a**.

Figure 3: *In silico* perturbation of GBM tumor cell scRNAseq data using an L1000-derived alisertib TCS predicts an NPC-like to MES-like tumor response confirmed *in vivo*

a. Histogram of single-cell alisertib TCS correlations (connectivities). Cells with negative correlation coefficients (blue) are predicted to be sensitive to alisertib, while cells with positive correlation coefficients (red) are predicted to be resistant to alisertib. **b.** Hierarchy plot of patient GBM tumor cells colored by predicted sensitivity ($p < 0$) or resistance ($p > 0$) to alisertib. **c.** Bar plot depicting mean shift in proportions of cells in resistant vs. sensitive populations within individual patient tumors. Error bars depict the standard error of the mean percentage differences of individual patient tumors. **d.** Schematic of *in vivo* experiments. **e.** UMAP plot of pre-filter single-cell transcriptomes colored by treatment with either alisertib or DMSO vehicle control. **f.** UMAP of captured cells from orthotopic xenografts colored by percent alignment to the human transcriptome (hg19). **g.** Dot plot of pass-filter, human xenograft single-cell transcriptome expression of Neftel et al. signatures, grouped by assigned transcriptional state identity as assigned based on predominant transcriptional state module expression. **h.** Two-dimensional hierarchical representation of GBM22 xenograft cells' relative enrichment scores for GBM cell transcriptional state modules. Cells are colored by assigned transcriptional state identity. **i.** Bar plot of

mean enrichment shift of transcriptional state signatures in alisertib-treated xenograft cells normalized to DMSO vehicle-treated xenograft cells. Error bars represent 95% confidence interval. (*Wilcoxon with Benjamini-Hochberg correction: *p-adjusted = 8e⁻⁹; **p-adjusted = 1.1e⁻⁶⁰; ***p-adjusted = 3.2e⁻²³⁸; NS p-adjusted = 0.79*) **j.** Alluvial plot depicting shift in relative proportion of transcriptional state identities in alisertib and DMSO vehicle control treated xenografts. **k.** Scatterplot of L1000 small molecule TCS's depicting predicted correlation shift (log₂FC) vs. observed correlation shift (log₂FC) in alisertib-treated xenografts. Differential small molecule correlations were calculated using *limma*.

The thought behind this analysis is now more complete and leads to subsequent predictions of potential synergistic combinations based on a reference small molecule, where the reference molecule connectivity divides single-cell input into resistant and sensitive populations, used in the calculation of an scFOCAL-based combination score discussed in more detail further down in this rebuttal. Importantly, we have revised our figures to reflect the reasoning behind this analysis more clearly, displaying our *in silico* prediction vs *in vivo* results in the same figure as our predictions and experimental data. Further, we have made a strong effort to better explain the significance of this approach and the figure in the text. Also importantly, additional analysis and figures demonstrate the utility of this scoring by effectively ranking the synergy of combinations as seen *in vitro*, and by identifying a novel combination therapy which is more effective than either monotherapy alone *in vivo*.

Figure 4a. Diagram of workflow for identifying CT-179 combinations using scFOCAL and cell-drug connectivity.

Minor: The GitHub repository for the manuscript is currently empty, I would suggest sharing it with the reviewers during the review process. For ISOSCELES Shiny application some tutorial dataset would be helpful.

We thank the reviewer for pointing this out. The GitHub repository for the scFOCAL application itself contains the scFOCAL R package source code and is available at <https://github.com/AyadLab/scFOCAL>. Within the readme file for this git, there are links to download tutorial datasets for use with the application. We have also since deposited all processing pipelines, including those for analyses performed during revision, within <https://github.com/AyadLab/scFOCAL-dataProcessing>. While the application is available on

shinyapps.io, we suggest installing it as an R package locally as per the readme file in the GitHub repository, as analysis with the tutorial dataset will run more efficiently, and shinyapps.io limits the needed memory usage.

README

CONTRIBUTORS 3 FORKS 0 STARS 0 ISSUES 0 OPEN LICENSE NOT SPECIFIED LINKED IN

scFOCAL

(s)ingle-(c)ell (F)ramework for (O)mics-(C)onnectivity and (A)nalysis via (L)1000

Explore the docs »

Report Bug · Request Feature

▼ Table of Contents

- 1. About scFOCAL
- 2. The scFOCAL R Package
 - Prerequisites
 - Installation
- 3. Usage
- 4. Contributing
- 5. License
- 6. Contact

No packages published
Publish your first package

Contributors 3

- RobertKSuter Robert K Suter
- amcgrew Anna McGrew Jermakowicz
- MatthewDAntuono

Languages

R 100.0%

Screenshot of the scFOCAL GitHub README

Reviewer 2: The manuscript describes application of disease signature reversal to the transcriptional profiles of single cells as a prediction of drug-sensitive and resistant populations in glioblastoma tumors. Based on scRNAseq from 11 GBM cases (a mix of primary and recurrent cases), the authors derive disease signatures for each of four GBM transcriptional cell states. The reversal of these disease signatures in L1000 small molecule transcriptomic dataset is then calculated to predict which subpopulation of GBM cells could be more sensitive or resistant to specific drugs. Conceptually, this is a very interesting approach, which could help identify drug combinations or sequences in which drugs should be combined to more effectively target all subpopulations in GBM. However, the current manuscript provides only a single example of a drug for which the computational analysis was performed and only a single PDX for experimental validation. It is thus unclear how robust is the ISOSCELES platform.

Specific comments: Minor comments:

Fig. 3 – would supervised clustering based on disease signature reversal specific to each cell state show significant differences between states? In other words, is L1000 dataset integration with expression data helpful as a discovery tool?

Regarding the reviewer’s comment on supervised clustering, we now show in new **Figure 2d-h** that clustering cells on their scFOCAL calculated drug connectivities is, in fact, very interesting. Importantly, we show that even in the context of only 63 FDA-approved oncology drugs, we can cluster and stratify cells in a way that reflects not only Neftel et al transcriptional state, but also other features such as differentiation state, and not surprisingly also by cell cycle. Importantly, we also demonstrate that in the context of these FDA-approved molecules, they are clustered by their connectivities to GBM tumor cells in such a way that reflects their respective mechanisms of action. (**Supplemental Figure S2**). Lastly, we demonstrate in **Figure 2h** that alisertib represents an FDA-approved (Orphan approval) compound which has significantly lower connectivity to NPC-like tumor cells in our dataset relative to the other states, as determined using *limma*.

Figure 2: Integration of single-cell expression and small molecule L1000 TCS signatures permits clustering of both compounds and cells by reversal of GBM cell transcriptional state-specific disease signatures. a. Schematic of single-cell sensitivity and resistance scoring. **b.** Correlation matrix depicting similarities of L1000 small molecule TCSs by their connectivity to all individual cells within our single-cell atlas. Row annotations depict compound TCS scores for the reversal of AC-, MES-, NPC-, and OPC-like disease signatures. **c.** Network plot of select L1000 small molecules colored by mechanism of action. Connections indicate a Pearson’s $\rho > 0.7$ between small molecules by their calculated

Fig. 4g – Is the change in percent composition significantly different after Alisertib treatment? Please provide the statistical test p-value.

We thank the reviewer for this important comment. As individual xenograft samples with shared treatment were pooled together, we are unable to perform a statistical significance assessment in regards to the percent composition before and after treatment. However, at the single-cell level, we can assess the shifts in relative Neftel et al state enrichment by comparing the enrichment scores for alisertib-treated tumor cells normalized to the mean enrichment scores of DMSO-treated tumor cells. Using a Wilcoxon signed-rank test and Benjamini-Hochberg multiple testing correction, we demonstrate significant enrichment shifts in AC-like, MES-like, and NPC-like module expression. Notably, OPC-like module enrichment was not significantly altered by alisertib treatment *in vivo*. These statistics have been added to **Figure 3i** as pictured below, and the prior panel depicting percent composition change as a bar plot has been removed in favor of the alluvial barplot in **Fig 3j**.

Figure 3i: Bar plot of the mean enrichment shift of transcriptional state signatures in alisertib-treated xenograft cells relative to the mean enrichments of DMSO vehicle-treated xenograft cells. Error bars represent the 95% confidence interval. (Wilcoxon with Benjamini-Hochberg correction performed for each state's normalized shift vs 0 (steady-enrichment): * p -adjusted = $8e^{-9}$; ** p -adjusted = $1.1e^{-60}$; *** p -adjusted = $3.2e^{-238}$; NS p -adjusted = 0.79) **Figure 3j:** Alluvial plot depicting shift in relative proportion of transcriptional state identities in alisertib and DMSO vehicle control-treated xenografts.

In the example provided, it is mentioned that several drugs are predicted to target the Alisertib-resistant population. It would be important to demonstrate if these predictions are correct by testing these drugs on resistant populations *in vitro* or combination treatment *in vivo*.

We thank the reviewer for this important perspective on the utility of the scFOCAL platform to identify combinations of small molecules that are predicted by the platform to target unique cell populations present within the same GBM tumor.

In our revisions, we've used the alisertib experiment to show that our platform predicts the relative sensitivities of alisertib resistant cells to different drugs (the compounds that we identify to be discordant with the predicted alisertib-resistant population from our patient dataset, also show increased discordance to cells in alisertib-treated tumors relative to vehicle treated tumors) to frame the potential to use our framework to design combinations. In summary, scFOCAL predicts differential drug connectivities occurring *in vivo*. We then go on to validate the significance of this demonstrating prediction and observed synergy correlation *in vitro* and *in vivo* using different small molecules. As our *in vivo* experiments with alisertib showed the OPC-like expression module was not significantly altered by alisertib, we hypothesized that the novel Olig2 inhibitor CT-179 would be predicted to target the OPC-like state using scFOCAL. Eventually, we demonstrate scFOCAL's ability to identify novel synergistic combinations *in vivo* using this inhibitor.

To this end, during revisions, we first refined the algorithm within the scFOCAL platform for identifying potentially synergistic combinations and visualization of its output. We applied this algorithm to the patient scGBM atlas used to generate combination scores for compounds used within the high-throughput synergy screen published in Houweling et al, 2023. This synergy screen was performed by the original authors on 25 different glioma stem cell lines in spheroid culture. Raw sum synergy measured by BLISS was compared to our scFOCAL combination index for tested molecules in combination with either erlotinib, lapatinib, pazopanib, or sunitinib. In both cases, overall, the combination index was significantly correlated with observed BLISS raw sum synergy. For the majority of cell lines tested, this combination index was very predictive of the relative synergy of the combinations tested. The most potent combinations in these instances were of those made of compounds with distinct TCS-connected cell populations.

Supplementary Figure S8: scFOCAL combination scoring predicts relative synergy of small molecule combinations *in vitro*. **a.** Stacked bar plots depicting the proportion of cells predicted to be sensitive or resistant across individual patient GBM tumors to reference small molecules erlotinib, lapatinib, pazopanib, and sunitinib, respectively. **b.** Heatmap of BLISS raw sum synergy of small molecule combinations obtained from Houweling et al., 2023. The top bar plot annotation depicts scFOCAL calculated combination score using labeled reference molecules depicted on the bottom annotation bar. Sidebar annotations depict Spearman's ρ and p-value between scFOCAL combination score and observed BLISS raw sum synergy for each cell line. **c.** Scatterplot of scFOCAL calculated combination score and mean BLISS raw sum synergy across all cell lines tested for small molecule combinations screened in Houweling et al., 2023 (Spearman's $\rho = 0.7$, p-value = 0.01). Points are colored by the reference molecule used for combination scoring. The shaded gray area around the line of best fit depicts the standard error.

Building on this analysis of prior *in vitro* screens, we moved to demonstrate an effective combination prioritized by our platform in a new *in vivo* experiment. Using a recently developed OLIG2 inhibitor, CT-179, a GBM cell line response signature was generated using bulk RNA sequencing (Figure 4a-b, Supplementary Fig. S9a-c). We demonstrate that this signature is enriched by CT-179 treatment in a dose-dependent manner, and then utilize this signature to predict relatively sensitive or resistant cell populations using our patient scRNAseq dataset (Figure 4c). Our predictions suggest that OPC-like cells would be sensitive, as would be

expected, and the AC-like population to be relatively more resistant (**Figure 4e**). Using our combination algorithm, we identified that compounds targeting EGFR and tubulin scored highly for synergy with CT-179 (**Figure 5a-c**). With this information, we demonstrate the combination of CT-179 and Depatux-M (a-EGFR /MMAF), an antibody drug conjugate combining the targeting of EGFR (over-expressed in AC-like cells) expressing cells with a tubulin poison, to increase survival *in vivo* relative to each monotherapy alone (**Figure 5d-e**).

Figure 4: The integration of a bulk-derived transcriptional response signature for the OLIG2 inhibitor CT-179 predicts targeting of OPC-like GBM cells. **a.** Diagram of workflow for identifying CT-179 combinations using SCFOCAL and cell-drug connectivity. **b.** Heatmap of GBM8 cells treated with vehicle or 200nM CT-179 for 24 hours. Columns represent biological replicates. **c.** Bar plot depicting the proportion of tumor cells within each patient tumor predicted to be sensitive (CT-179 response $p < 0$) or resistant (CT-179 response $p > 0$) to CT-179 treatment. **d.** Hierarchy plot of GBM tumor cells arranged by their relative expression of Neftel et al. states and colored by predicted sensitivity or resistance to CT-179 treatment. **e.** Violin and box plot of GBM cell CT-179 connectivity (ρ) grouped by assigned GBM cell transcriptional state.

Figure 5: An scFOCAL combination index predicts a synergistic combination of an OLIG2 inhibitor CT-179 with Depatux-M (ABT-414), an anti-EGFR antibody MMAF drug conjugate. **a.** Volcano plot depicting the results from limma-based differential drug connectivity between predicted CT-179 sensitive and resistant GBM cells. Patient ID was used as a covariate in the model design. **b.** Barplot of L1000 small molecule mean resistant cell connectivity (RCC) to predicted CT-179 resistant cells. Color depicts this same value. **c.** Scatterplot of L1000 small molecules plotted by resistant vs. sensitive differential connectivity and mean CT-179 resistant cell connectivity. Colors depict the calculated scFOCAL combination index, the product of multiplying the differential connectivity log₂FC values by the mean CT-179 resistant cell connectivity values for each molecule. Compounds highlighted in **(a)**, **(b)**, and **(c)** are varlitinib, an EGFR inhibitor, and indibulin and docetaxel, which act through inhibition of tubulin. **d.** Bioluminescence signal quantification of GBM6-eGFP-FLUC2 orthotopic xenograft tumors (means ± SD, Combo vs Vehicle: p.adj < 0.005 from Day 14, Combo vs. CT-179 monotherapy: p.adj < 0.05 from Day 11, Adjusted p-values from multiple t-tests with Holm-Šidák correction to control the family-wise error rate) **e.** Kaplan-Meier survival curves of mice bearing GBM6-eGFP-FLUC2 orthotopic xenografts treated with indicated therapies (n = 10 per group). MS: median survival. P-values determined using the Log-rank (Mantel-Cox) test.

Supplementary Figure S9: Generation of a GBM CT-179 transcriptional response signature. a. Volcano plot of differential expression between 200nM CT-179 and vehicle-treated GBM8 cells calculated via DESeq2. **b.** Barplot depicting gene ontology analysis of the filtered CT-179 transcriptional response signature. Bars are colored by the respective ontology used. Selected GO terms highlighted include ‘microtubule’, ‘microtubule binding’, ‘tubulin binding’, and ‘microtubule motor activity’. **c.** Barplot of singscore-calculated enrichment for the directional CT-179 transcriptional response signature in 200nM CT-179-treated, 100nM CT-179-treated, and vehicle-treated GBM8 cells after 24 hours. Significance bars depict p-value following pairwise t-tests (n = 3 per treatment group). **d.** Box and violin plots depicting the expression of EGFR and OLIG2 in patient GBM tumor cells grouped by assigned Neftel et al. GBM cell transcriptional state.

The scFOCAL figure (Fig. 5, Results, Proportion shift tab) suggests that there are two populations expanding as resistant – MES1 and AC. How can this be reconciled with results presented in Fig. 4? Experiments performed on another cell line or two could help determine whether these expansions depend on the initial fraction of the cell state in untreated tumor.

Thank you for bringing this to our attention. This figure was meant to be illustrative and was based on an outdated and sub-sampled testing dataset, prior to calculation of overall MES and NPC scores (i.e MES1/2, NPC1/2). We have updated **Figure 6** to illustrate scFOCAL analysis of the sample data included with the GEO release.

Figure 6: The scFOCAL framework is available for use as a shiny web application or an R package.

The current scFOCAL model is built on 11 datasets, of which 4 are recurrent GBM cases. Would the predictions be better if only primary cases were used? Would they look different for only recurrent cases? There is several publicly available datasets based on 10xGenomics platform that could be used to test this: ex. 16 patients in PMID: 32641768 , 10 patients in PMID: 35122077, 11 patients in PMID: 31901251 and many others.

Thank you for your comment. We sought to develop predictions regardless of newly diagnosed or recurrent tumor status, as many patients seek out trials upon tumor recurrence. As such, we had not tried to separate newly diagnosed from recurrent tumors since we did not have enough recurrent GBM patient tumors for these predictions. A single-cell atlas with an adequate number of paired longitudinal primary and recurrent samples would be required to adequately examine this further. However, such a dataset is not currently available. Our statistical analysis suggests we need at least 10 samples per group to identify differences, which is not available in our current dataset. Therefore, we will perform this analysis when more samples are available.

The overall robustness of the study would be significantly strengthened by additional PDXs tested for their Alisertib response. In a heterogeneous disease, such as GBM, a single cell line/PDX might not be representative.

We thank the reviewer for this important comment, and have since shifted the focus of our manuscript in response to this comment as well as others. To address this comment specifically, we have included additional analyses demonstrating our platform's performance in the context of different small molecules and perspectives.

To highlight the utility of our platform, we have first applied scFOCAL to publicly available scRNAseq data of patient GBM acute slice cultures treated with either DMSO or the HDAC inhibitor panobinostat, and showed that scFOCAL calculated panobinostat connectivity of single-cells correlates with panobinostat treatment *ex vivo* (Panobinostat TCS derived from the L1000 (Supp. Fig 7c)). Here we demonstrate that L1000-derived panobinostat connectivity calculated with scFOCAL is a predictor of whether a cell has been treated with panobinostat or not. Interestingly, in the original paper where this data originated, the authors identify that panobinostat treatment dramatically affected the tumor-associated myeloid cell population as well. scFOCAL cell-drug connectivity identifies this difference as well (Supp. Fig 7de).

Supplementary Figure S7: scFOCAL connectivity analysis with an L1000-derived panobinostat TCS separates vehicle-treated from panobinostat-treated cells in both neoplastic and myeloid cell populations. a. UMAP of 62,250 single-cell transcriptomes from patient-derived acute slice culture samples obtained from Zhao et al., 2021, treated with DMSO or the HDAC inhibitor panobinostat (0.2 μ M). **b.** UMAP of single-cell transcriptomes colored by discrete cell type. **c.** Heatmap showing z-scores of panobinostat-specific gene expression (vs. treatment naïve) for genes retained in the Panobinostat TCS across cell lines tested in the L1000 dataset. **d.** Violin plots showing panobinostat TCS connectivity

of Panobinostat-treated vs. DMSO-treated cells in neoplastic and myeloid populations, respectively (Wilcoxon rank sum test with continuity correction, p-value < 2.2e-16). **e.** Area-normalized kernel-density estimate (KDE) plots of scFOCAL calculated panobinostat TCS connectivity colored by treatment with DMSO or panobinostat, across neoplastic and myeloid cell populations, respectively.

We too recognize that this extremely heterogeneous cancer likely exhibits varying responses to different drugs depending on the cell line utilized or the patient from whom a tumor originated. Thus, we have focused more on the aspect of combination design, where input data will change and thus will affect predictions, and that scFOCAL represents a step towards personalized medicine, rather than towards the identification of a one-size-fits-all treatment. As such, we have included a separate analysis of a different small molecule predicted to target a different GBM cell state, with a different predicted resistant GBM cell state. Leveraging this, we identify a novel combination and demonstrate its increased efficacy relative to each component as monotherapy alone.

Taken together, even though we have not profiled additional PDXs with alisertib, we have included a variety of different use-cases and perspectives in support of our framework's capability that we believe best suit its intended purpose.

Please include a summary of the number of cells and gene count per cell to describe the single-cell datasets used in this study.

Thank you for addressing this important oversight. We have since added cell and detected gene counts within the manuscript for patient, xenograft, and newly analyzed acute slice culture scRNAseq datasets.

CytoTRACE-based assignment of malignant and non-malignant state should be briefly explained in the text to clarify what is presented in Fig. 1d.

Thank you for your comment. We have updated the methods with a brief explanation of CytoTRACE to clarify the importance of **Fig. 1d**, that the tumor cell population we identify within our dataset is de-differentiated relative to known non-tumor cells, and further that the tumor cell population is heterogeneous in the context of transcriptional diversity and thus developmental potential.

Reference to Fig.3a in the text is missing.

Thank you for bringing this to our attention. We have made sure that every figure, including supplementary figures, is mentioned and referenced within the text.

Reviewer 3: GBM therapy, preclinical models (Remarks to the Author): In this manuscript, the authors present an interesting general approach: the development of an algorithm that allows to predict sensitivity of glioblastoma cell subpopulations (i.e., NPC-, OPC-, MES- and AC-like cells) to certain drugs and drug classes. Tumor heterogeneity is a formidable

challenge in the treatment of glioblastoma, and a better idea of which transcriptional tumor cell subpopulations respond to which drugs/classes can help to develop tailored multi- or oligo-drug strategies; potentially even identifying one drug that is able to target all, or at least the most important, tumor cell subpopulations. Having said that, this manuscript falls short of providing evidence that the computational *in silico* approach presented here is of any practical (biological / therapeutic / mechanistic) value.

The investigation of one single drug in one single cell line *in vivo* demonstrating one major change - a shift towards MES-like tumor cell subpopulation - is no evidence for the utility of this entire approach. A shift towards MES-like states is well known for many therapeutic interventions now, and far away from any proof of a specific alteration induced.

We thank the reviewer for their important comment. We agree that as this tool was developed, the understanding of a common proneural-to-mesenchymal transcriptional response in glioma has been increasingly characterized. In line with comments from another reviewer, we have addressed this issue by identifying different small molecules which are predicted to have varying responses to that predicted and observed in response to alisertib. Importantly, not all proneural to mesenchymal programs are the same, and differing NPC-like and MES-like states have already been characterized. While not within the scope of this paper, more mechanistic studies on the proneural to mesenchymal response in GBM need to be performed.

However, to address this comment, we have dramatically shifted the focus of the paper more towards our platform's ability to effectively identify more effective combinations by identifying compounds that target distinct GBM cell transcriptional states present within the same tumor. Additionally, we have included an additional analysis demonstrating the predictive value of scFOCAL cell-drug connectivity utilizing different small molecules.

First, to highlight the utility of scFOCAL in predicting heterogeneous cell population response outside of a proneural to mesenchymal shift, we demonstrate that cell-drug connectivity is predictive in the context of the HDAC inhibitor panobinostat using publicly available data of panobinostat-treated patient GBM acute slice cultures. Here, we also demonstrate the utility of scFOCAL to predict dynamics of the tumor-associated macrophage population within these same tumors, as described by the authors of the data's originating manuscript (Zhao et al, 2021). Panobinostat TCS connectivity very effectively separates DMSO and panobinostat-treated tumor cells and macrophages on a probabilistic distribution (**Supp. Fig 7de**).

Supplementary Figure S7: scFOCAL connectivity analysis with an L1000-derived panobinostat TCS separates vehicle-treated from panobinostat-treated cells in both neoplastic and myeloid cell populations. **a.** UMAP of 62,250 single-cell transcriptomes from patient-derived acute slice culture samples obtained from Zhao et al., 2021, treated with DMSO or the HDAC inhibitor panobinostat (0.2 μ M). **b.** UMAP of single-cell transcriptomes colored by discrete cell type. **c.** Heatmap showing z-scores of panobinostat-specific gene expression (vs. treatment naïve) for genes retained in the Panobinostat TCS across cell lines tested in the L1000 dataset. **d.** Violin plots showing panobinostat TCS connectivity of Panobinostat-treated vs. DMSO-treated cells in neoplastic and myeloid populations, respectively (Wilcoxon rank sum test with continuity correction, p-value < 2.2e-16). **e.** Area-normalized kernel-density estimate (KDE) plots of scFOCAL calculated panobinostat TCS connectivity colored by treatment with DMSO or panobinostat, across neoplastic and myeloid cell populations, respectively.

Next, we have developed a combination scoring approach within our platform, and demonstrate its ability to predict the relative synergy of small molecule combinations using publicly available synergy screen data obtained from Houweling et al. By effectively performing *in silico* perturbation studies using our patient scRNAseq dataset and performing differential connectivity analysis between predicted reference-drug-sensitive and -resistant populations, we derive a combination score that positively correlates with BLISS synergy *in vitro* in a large panel of GBM cell cultures.

Supplementary Figure S8: scFOCAL combination scoring predicts relative synergy of small molecule combinations *in vitro*. **a.** Stacked bar plots depicting the proportion of cells predicted to be sensitive or resistant across individual patient GBM tumors to reference molecules erlotinib, lapatinib, pazopanib, and sunitinib, respectively. **b.** Heatmap of BLISS raw sum synergy of small molecule combinations obtained from Houweling et al., 2023. The top bar plot annotation depicts scFOCAL calculated combination score using labeled reference molecules depicted on the bottom annotation bar. Sidebar annotations depict Spearman's ρ and p-value between scFOCAL combination score and observed BLISS raw sum synergy for each cell line. **c.** Scatterplot of scFOCAL calculated combination score and mean BLISS raw sum synergy across all cell lines tested for small molecule combinations screened in Houweling et al., 2023 (Spearman's $\rho = 0.7$, p-value = 0.01). Points are colored by the reference molecule used for combination scoring. The shaded gray area around the line of best fit depicts the standard error.

We wanted to demonstrate that our approach is useful for predicting drugs targeting different cell populations in GBM tumors. For this reason, we included *in vivo* studies with the Olig 2 inhibitor CT179. CT-179 is predicted to target OPC-like cells and yield an AC-like resistant cell population. Through the prioritization of EGFR and tubulin targeting molecules by our combination prioritization strategy, which we previously included in our original submission as Supplementary Figure S6, we validate our predictions from a CT-179 response signature by demonstrating synergy *in vivo* with CT-179 and DepatuxM, an anti-EGFR MMAF (tubulin inhibitor) antibody-drug conjugate (**Fig. 4, Fig 5, Fig S9**). Importantly, our alisertib experiment still fits the altered narrative of this manuscript through demonstration that *in silico* perturbation of our patient GBM atlas accurately predicts differential connectivities to other drugs between the vehicle and alisertib treated PDX tumors (**Fig 3k** above, **Supplementary Fig. S6**), which led us to investigate the potential for our framework to prioritize synergistic combinations in our original submission.

Supplementary Figure S6: In silico perturbation and predictive differential connectivity analysis models the drug-connectivity perturbation response to alisertib *in vivo*. **a-h.** Scatterplots of L1000

small molecule TCSs within individual compound classes depicting scFOCAL-predicted correlation shift (\log_2FC) vs. observed correlation shift (\log_2FC) in alisertib-treated xenografts. Differential small molecule correlations were calculated using *limma*.

They need to study the response of all 4 subpopulations to a panel of drugs experimentally; subpopulations can be studied by enrichment/depletion analyses, better individually after FACS sorting or, ideally, and something this reviewer would strongly recommend, using in vivo reporter systems or bar code systems that allow to distinguish all 4 subpopulations. This should be done in state-of-the-art in vitro assays.

We thank the reviewer for these suggestions. We have steered the focus of our manuscript towards our platform's ability to effectively rank more effective combinations. To do this, we have focused on analyzing existing datasets that utilize L1000-tested compounds, which supported our novel platform scFOCAL, and performed new experiments using the OLIG2 inhibitor CT-179, generating a CT-179 transcriptional response signature, which we used to demonstrate *in vivo* that scFOCAL can identify novel synergistic combinations.

After that, the authors need to demonstrate in vitro and in vivo that the most promising compound (mix) is indeed providing a therapeutic advantage compared to other approaches.

We have mined previously published data for panobinostat in slice cultures, and we demonstrate that the scFOCAL predictions are consistent with what was observed by single-cell analysis *ex vivo*. These studies further support that our pipeline predicts the cells that are targeted pharmacologically in GBM, and that drug-cell connectivity is a predictive value of how cells respond to different treatments. As suggested, we have also extended our findings to develop a scFOCAL combination index to score combinations for potential synergy. Using mined synergy screen data from Houwelling et al, 2023, we score small molecule combinations using scFOCAL, and found that our predictions correlate with relative synergy results across 24 GBM cell lines in spheroid culture. Having validated our combination index using *in vitro* data, we moved to identify a novel combination using the OLIG2 inhibitor CT-179, and demonstrated the efficacy of our prioritized combination *in vivo*.

More minor point: the presentation of the results and figure legends is often confusing and should be significantly improved. All figure panels should be cited in the order of appearance; it should be explained in the results section what they actually show; and the reader needs to get more explanatory information there, too.

We thank the reviewer for these suggestions and have now modified the order of the figures to be consistent with the order of appearance, in addition to the major overhaul of all of the figures included. We have also explained in depth what the results show, as suggested in the figure legends and results.

We thank the reviewers for their positive assessment of our revised manuscript. We have now addressed Reviewer 2's concerns point-by-point in a second revision. We hope our manuscript is now acceptable for publication at *Nature Communications*.

REVIEWER COMMENTS:

- **Reviewer #1:**
 - “Thank the authors for answering my questions, and also for the rich additional analysis. I think the synergy prediction strongly increased the importance of this paper.”

- **Reviewer #3:**
 - “The authors have responded well to many of my comments and have now provided a better proof-of-principle that their novel cell state-based approach can indeed help to identify new therapeutic combinations.”

- **Reviewer #2 (Comments Addressed)**
 - **Comment #1:** “This is a revised manuscript describing a new computational tool for predicting drug sensitivity at the single-cell level. The main point raised in the prior round of review was the robustness of the prediction, as limited experimental data were shown, and the question about the ability to predict combinations of drugs to target the subpopulations. The presented approach to identify drug combinations is highly warranted. However, the scFOCAL wouldn't have been needed to come up with a combination of OLIG2i and EGFR-MMAF, since both targets are what distinguish the transcriptional cell states in GBM. Docetaxel and varlitinb don't seem to be the top hits in the connectivity analysis. It would be important for the readers to understand how to prioritize hits that may be selected when they apply scFOCAL to their own dataset.”
 - **Comment #1 Response:** We thank the reviewer for their positive assessment of the need for our platform to predict combinations in an unbiased manner at the single-cell level. Regarding whether scFOCAL would have been needed to come up with a combination of OLIG2i and α -EGFR-MMAF, we would like to point out that prior to our findings, no other group has predicted and demonstrated that the combination of an OLIG2i and α -EGFR-MMAF produces a survival benefit in preclinical models of GBM. Our work does exactly that and puts forth a novel clinical candidate, OLIG2i CT-179, as a possible combination therapy in GBM. These findings are essential to clinical trial designs that will emanate from this work. Therefore, our findings are significant and important for the medical and scientific GBM community.

A neuro-oncology expert may indeed identify that this combination makes sense in the context of GBM cell transcriptional states. Uniquely, scFOCAL identifies that OLIG2 inhibition may sensitize cells to

drugs acting on tubulin, simultaneously identifying that these sensitized cells likely also express EGFR in a single quantitative measure, our combination score. The results of our *in silico* combination screening, taken together in context with our identification of EGFR-expressing AC-like GBM cells as the most concordant with our CT-179 response signature, motivated us to validate the combination of CT-179 and Depatux-M *in vivo* over exploring alternative targets.

Importantly, Depatux-M does not necessarily inhibit EGFR signaling, but utilizes EGFR expression to guide a tubulin poison to the target cells. While varlitinib, an EGFR inhibitor, scores highly for our combination index, this is mainly contributed to by its corresponding logFC differential in connectivity, suggesting that the connectivity between CT-179 sensitive and resistant cells differs, and that varlitinib is predicted to target different cells than CT-179. By contrast, indibulin's high combination index is mainly contributed to by the magnitude of its mean discordance with CT-179-resistant cells. Combined with the interpretation of the scFOCAL prediction of AC-like cell state resistance, Depatux-M, leveraging this approach to kill the cell, while homing to EGFR expressing cells (regardless of EGFR activation), makes sense in the context of the individual factors comprising the scFOCAL combination index.

In reference to the observation that docetaxel and varlitinib are not the top hits in our connectivity analysis, we would like to point out that they are all within in the top 31 candidates from 1674 molecules we initially began with (Top 1.8%), from the 1382 annotated as clinically tested by the repurposing hub (Top 2.25%), and from the 411 passing filter for being sensitized by CT-179 *in silico* and for being discordant with CT-179-predicted-resistant cells (Top 7.5%). Depatux-M is a more attractive therapeutic than either alone due to its status as an antibody drug conjugate targeting EGFR. Further, Depatux-M has been utilized in a Phase III clinical trial for glioblastoma, which demonstrated extension of progression-free survival (PMID: 35849035), indicating its safety and BBB penetration. Lastly, our alisertib studies demonstrated the accuracy in predictions of tubulin polymerization inhibitors, lending to further confidence in prioritizing indibulin and docetaxel.

Regarding how an individual reader can prioritize hits that may be selected when they apply scFOCAL to their own dataset, we provide a combination score that can rank various compounds. This combination score must be considered in the context of their disease of interest and in the biology underlying heterogeneity within that specific disease. Further studies will be required to determine the possible cut-offs for scores to predict combinations that are validated through *in vitro* and *in vivo* combination studies. We now add to the limitations of the study this important point, that although our combination index provides a predicted measure of synergy of compounds, as a function of how broadly the combination targets a heterogeneous tumor cell landscape, the identification of effective combinations further requires an iterative process, where data and information from the scientific literature must be considered. Importantly, our manuscript demonstrates the presence and utility of a biologically relevant cell-and-drug transcriptional connectivity signal, of which further developments can be applied to further strengthen its application.

- **Comment #2: “CytoTRACE should be better described in the text, as it is unclear what it represents in Fig. 2f&g.”**
- **Comment #2 Response:** In the discussion section, we have further explained our interpretation of drug connectivity clustering of cells exhibiting patterns with tumor cell CytoTRACE scores, elaborating on the interpretation that cells clustering on drug connectivity correlating with different relative CytoTRACE scores indicates that tumor cells in different differentiation states exhibit unique sensitivities from more differentiated cell states. Cells with a higher CytoTRACE score have a higher transcriptional diversity, and thus represent more stem-like de-differentiated tumor cell states. In our new **Supplementary Fig. S3b**, it can be better seen that these CytoTRACE scores are correlated with NPC-like GBM cell transcriptional states.

- **Comment #3: “Line 168: 'Within each individual, cycling cells seem to separate into a unique drug connectivity state’. Is this true for all patients in this cohort or only for the two tumors presented in Fig. 2?’”**
- **Comment #3 Response:** We thank the reviewer for this important point. In response, we have added a new supplementary figure (**Supplementary Fig. S3**). This figure first presents the individual clustering heatmaps for each of the 11 patients. We then performed hierarchical clustering on the resulting cell groups using three key features: **1) proportions of Nefel GBM states, 2) cell cycle phase proportions, and 3) the mean CytoTRACE score**. This analysis reveals that the patient clusters organize into distinct, high-level metaclusters. We have added further comments on this within our revised manuscript and have removed the statement in question. While a proliferative meta-cluster is apparent, there are also variations in cycling proportion within metaclusters.

Supplementary Figure S3: Cell-drug connectivity using FDA-approved oncology drugs clusters patient GBM tumor cells reflecting patterns of GBM cell transcriptional state. **a)** Correlation matrices depicting pairwise Spearman correlations of single GBM tumor cells from each patient’s tumor cells. Column annotations depict each cell’s CytoTRACE score, cell cycle phase, and dominant Neftel et al. transcriptional state. **b)** Heatmap depicting hierarchical clustering of all clusters k from all 11 patients on the relative proportions of cells within each GBM cell transcriptional state, cell cycle phase, and mean CytoTRACE score.

- **Comment #4:** “The authors focus on alisertib, based on previous connection to MES cell state. There are a number of other candidates, including alpelisib and afatinib, which have highly differential connectivity between cell states. This should be clarified.”

- **Comment #4 Response:** We thank the reviewer for this important request, as other small molecules may be of interest to other researchers. Our prior unpublished work shows that alisertib monotherapy elicits a significant increase in survival time in orthotopic xenografts. This is consistent with what was previously reported about alisertib (PMID:27816996). Thus, we were interested in characterizing the eventual development of resistance and tumor growth despite treatment with alisertib. Further, our prior publications indicate that alisertib is well tolerated in vivo and does get into the brain. While we agree that alpelisib and afatinib are of interest, we do not have in vivo data to support their efficacy or brain penetrance in our hands. We can pursue these studies in the future and comment in the Discussion that similar molecules should be analyzed further.

Survival proportions: Survival of DMSO_v_Alisertib

Rebuttal Figure 1: Kaplan-Meier survival curve for GBM22-bearing mice treated with DMSO or alisertib. (unpublished)

- **Comment #5: Fig.3a.** Predicted sensitive vs resistant populations are divided right at 0 point for TCS correlation coefficient. Would the differences be more striking if top and bottom quartiles were used? With the current split, in Fig.3b the distribution looks very similar.
- **Comment #5 Response:** We thank the reviewer for this important question, and have added additional analysis to **Supplementary Figure S4** to address this. To demonstrate that the scFOCAL prediction for relative sensitivity or resistance to alisertib in the context of Neftel et al. states is robust, we have iteratively re-created **Figure 3c** across different quantiles of alisertib connectivity (i.e., top 5% vs bottom 5%, top 10% vs bottom 10%, through to the top 45% vs the bottom 45%). Within each patient, the proportion change of each cell state was compared using Wilcoxon rank-sum tests and Holm multiple comparisons correction. At each different quantile comparison, the differences in proportion are significant between MES-like and NPC-like populations. At the 5% comparison, MES and OPC-like proportion shifts are also statistically significant.

Supplementary Figure S4: Interrogation of an alisertib TCS using single-cell RNA sequencing indicates the predicted sensitivity of an NPC-like GBM cell state. **a.** Diagram describing interrogation of the L1000-derived alisertib TCS using GBM cell state markers as in **(b)** or by discordance ratio with GBM cell state disease signatures as in **(c)**. **b.** Heatmap depicting differential expression of cell state markers that overlap with alisertib TCS genes. Cell color indicates the relative log₂FC of each gene, and black heatmap cells indicate a gene was not differentially expressed by a given cell state. Row annotation depicts the magnitude and direction of gene expression changes of the alisertib TCS. **c.** Box plot of cell state disease signature discordance to the alisertib TCS by individual patient tumor. Cell state disease signatures were calculated using MAST to compare cells within each GBM cell state to non-neoplastic cell types within each individual. **d.** Bar plot depicting the pairwise post-hoc $-\log(p.\text{adj})$ values of proportion change for each state combination as calculated from main Figure 3c. **e-m.** Histograms of patient GBM tumor cells colored by assigned alisertib TCS quantile-based sensitivity or resistance, paired with corresponding box plots of predicted proportion change for each Neftel et al. GBM cell transcriptional state. For each quantile cut-off, the change in proportion was calculated by subtracting the proportion in the sensitive group from that of the resistant group. Post-hoc pairwise comparisons of relative state shifts were performed with the rstatix package using Wilcoxon rank-sum tests, and the resulting p values were adjusted for multiple comparisons using the Holm method (* $p.\text{adj} < 0.05$, ** $p.\text{adj} < 0.005$).

- **Comment #6: Fig. 3j.** Is the increase in MES and decrease in NPC after alpelisib treatment statistically significant? The changes are not massive, so statistical test would be helpful.
- **Comment #6 Response:** We thank the reviewer for this important comment. The Neftel et al. state identities of the tumor cells are assigned based on the state most enriched in that cell by singscore npc enrichment using the original 50 gene signatures. Because for this experiment, individual tumors from 3 mice per treatment were pooled into single GEM captures (10X), we are unable to perform statistical tests on proportions of cells. However, in **Figure 3i**, depicting the normalized shift in each state's enrichment utilizing individual cells as datapoints, we are able to perform statistics and demonstrate that the NPC-like signature depletion and MES-like signature enrichment are statistically significant, while

the change for OPC-like enrichment does not reach significance. Panel **3j** is another perspective of **3i**, depicting changes in the predominantly expressed signatures of tumor cells, calculated from the significantly changed enrichment values depicted in **Fig. 3i**.

Fig. 3i. Bar plot of mean enrichment shift of transcriptional state signatures in alisertib-treated xenograft cells normalized to DMSO vehicle-treated xenograft cells. Error bars represent 95% confidence interval (Wilcoxon with Benjamini-Hochberg correction: *p-adjusted = $8e^{-9}$; **p-adjusted = $1.1e^{-60}$; ***p-adjusted = $3.2e^{-238}$; NS p-adjusted = 0.79). **Fig. 3j.** Alluvial plot depicting shift in relative proportion of transcriptional state identities in alisertib and DMSO vehicle control-treated xenografts.

- **Comment #7: Fig. 3k.** The correlation between observed and predicted shifts is weak overall, yet for some classes of compounds they work well (Fig. S6). It might be worth to put S6 panels in the main figure, since they represent a more meaningful result. Is there any connection between these drug classes and action or transcriptional consequences of alisertib treatment?
- **Comment #7 Response:** We thank the reviewer for this important suggestion. We have now moved the panels for topoisomerase inhibitors, NRT (nucleoside reverse transcriptase), and serotonin receptor agonists from **Supplementary Fig. S6** to the main figure, **Fig. 3k**, as suggested. We chose to further highlight these compound classes to demonstrate the predictive power to predict compounds with decreased cell discordance following alisertib treatment, such as the majority of topoisomerase inhibitors, but also to show the power to predict compounds with increased discordance against tumor cells, such as NRT and select serotonin receptor agonists. In the manuscript, we have added a discussion of the relationship between topoisomerase inhibition and reduced expression of Aurora Kinase A, the target of alisertib. Further, we have added a comment on the potential for future investigation of NRT and serotonin receptor agonists, as we have previously published on the potential of repurposing psychiatric drugs for brain cancers, particularly due to their known blood-brain barrier penetrance.

Figure 3: In silico perturbation of GBM tumor cell scRNAseq data using an L1000-derived alisertib TCS predicts an NPC-like to MES-like tumor response confirmed in vivo. a. Histogram of single-cell alisertib TCS correlations (connectivities). Cells with negative correlation coefficients (blue) are predicted to be sensitive to alisertib, while cells with positive correlation coefficients (red) are predicted to be resistant to alisertib. **b.** Hierarchy plot of patient GBM tumor cells colored by predicted sensitivity ($\rho < 0$) or resistance ($\rho > 0$) to alisertib. **c.** Bar plot depicting mean shift in proportions of cells in resistant vs. sensitive populations

within individual patient tumors. Error bars depict the standard error of the mean percentage differences of individual patient tumors. **d.** Schematic of in vivo experiments. **e.** UMAP plot of pre-filter single-cell transcriptomes colored by treatment with either alisertib or DMSO vehicle control. **f.** UMAP of captured cells from orthotopic xenografts colored by percent alignment to the human transcriptome (hg19). **g.** Dot plot of pass-filter, human xenograft single-cell transcriptome expression of Neftel et al. signatures, grouped by assigned transcriptional state identity, as assigned based on predominant transcriptional state module expression. **h.** Two-dimensional hierarchical representation of GBM22 xenograft cells' relative enrichment scores for GBM cell transcriptional state modules. Cells are colored by assigned transcriptional state identity. **i.** Bar plot of mean enrichment shift of transcriptional state signatures in alisertib-treated xenograft cells normalized to DMSO vehicle-treated xenograft cells. Error bars represent 95% confidence interval (Wilcoxon with Benjamini-Hochberg correction: *p-adjusted = 8e-9; **p-adjusted = 1.1e-60; ***p-adjusted = 3.2e-238; NS p-adjusted = 0.79). **j.** Alluvial plot depicting shift in relative proportion of transcriptional state identities in alisertib and DMSO vehicle control-treated xenografts. **k.** Scatterplots of L1000 small molecules and select subsets of compound classes' TCSs depicting predicted correlation shift (log₂FC) vs. observed correlation shift (log₂FC) in alisertib-treated xenografts. Differential small molecule correlations were calculated using limma. R (ρ) values determined using the Pearson correlation.

- **Comment #8: The Panobinostat response analysis seems a bit disconnected from the rest of the manuscript. Since this is an scRNAseq dataset, are the authors able to see changes in Neftel et al. cell states in the cultured slices?**
- **Comment #8 Response:** We thank the reviewer for this important request to increase the relevance of the meta-analysis of the acute slice culture treatment data included from Zhao et al. To do this, we have further analyzed the dataset in the context of Neftel et al. transcriptional states, and show that the pattern of shifts in transcriptional states induced by panobinostat treatment is also correlated with increases in panobinostat TCS connectivity. To do this, we have split the data on the panobinostat TCS connectivity quartiles, and analyzed the shift in proportions of cells predominantly within each of the 4 transcriptional states. (**Supplementary Fig. S4**). We directly compare these predicted shifts to those observed in the actual treatment groups across all 5 individual patient sources. While the proportion shifts between treatment groups do not reach statistical significance, a clear trend is apparent, which panobinostat TCS connectivity also identifies and directly correlates with. Further, the inclusion of this dataset and meta-analysis within the supplementary material is warranted due to the perceived targeting of AC-like cell states by panobinostat, unique from both alisertib and CT-179, which were found to be most transcriptionally discordant with NPC-like and OPC-like cells, respectively.

Supplementary Figure S7: scFOCAL connectivity analysis with an L1000-derived panobinostat TCS separates vehicle-treated from panobinostat-treated cells in both neoplastic and myeloid cell populations. **a.** UMAP of 62,250 single-cell transcriptomes from patient-derived acute slice culture samples obtained from Zhao et al., 2021, treated with DMSO or the HDAC inhibitor panobinostat (0.2 μ M). **b.** UMAP of single-cell transcriptomes colored by discrete cell type. **c.** Heatmap showing z-scores of panobinostat-specific gene expression (vs. treatment naïve) for genes retained in the Panobinostat TCS across cell lines tested in the L1000 dataset. **d.** Violin plots showing panobinostat TCS connectivity of Panobinostat-treated vs. DMSO-treated cells in neoplastic and myeloid populations, respectively (Wilcoxon rank sum test with continuity correction, p -value $< 2.2e-16$). **e.** Area-normalized kernel-density estimate (KDE) plots of scFOCAL

calculated panobinostat TCS connectivity colored by treatment with DMSO or panobinostat, across neoplastic and myeloid cell populations, respectively. **f.** Barplots of cell proportion for each GBM cell transcriptional state, with cells split based on panobinostat TCS connectivity quartiles (high vs low). **g.** Barplots of cell proportion for each GBM cell transcriptional state, comparing DMSO vehicle and panobinostat-treated tumors **h.** Scatterplot of proportion shifts predicted from Q1-Q3 comparisons, plotted against proportion shift between treatment groups. R value determined using Spearman correlation.

-
- **Comment #9: “Docetaxel and varlitinb don’t seem to come up as top hits in the connectivity analysis. Could the authors comment on the top hits and why they were not pursued for experimental validation?”**
- **Comment #9 Response:** We thank the reviewer for this important comment. In response, we would first like to clarify that these 3 compounds highlighted behind the interest of Depatux-M, namely indibulin, docetaxel, and varlitinib, are prioritized within the top 7% of the 411 molecules passing both the negative-logFC and negative mean resistant cell connectivity filters (Indibulin: 16/411 (top 3.9%); Docetaxel: 21/411 (top 5.1%); Varlitinib: 31/411 (top 7.6%)). Prior to this filtering, these 411 compounds are drawn from a pool of 1382 small molecules selected for use in clinical trials as annotated in the repurposing hub (Indibulin: 16/1382 (Top 1.2%); Docetaxel 21/1382 (Top 1.5%); Varlitinib 31/1382 (Top 2.3%)). Importantly, the coloring in **Figure 5c** depicts the scFOCAL combination index, which we leverage for prioritizing. While other top hits are definitely interesting, we sought to prioritize a combination that would demonstrate that targeting distinct GBM cell states simultaneously improves survival. Depatux-M, harboring qualities of multiple scFOCAL prioritized hits, has been utilized in phase III trials, is safe, and extended PFS in glioblastoma (PMID 35849035). To this end, we have added a further detailed discussion of our focus on Depatux-M. Lastly, we have made sure to include a table of CT-179-resistant vs sensitive logFC, mean CT-179-resistant cell connectivity (MRC), and combination index values in the supplementary data, within Supplementary Data 6.
- **Comment #10: Fig. 2b – scale for disease signature reversal is missing**
- **Comment #10 Response:** We thank the reviewer for catching this important omission. We have since added the scale for each individual Neftel et al. state discordance ratio for the annotation of the heatmap in **Fig. 2b**.

Updated Fig 2b. Correlation matrix depicting similarities of L1000 small molecule TCSs by their connectivity to all individual cells within our single-cell atlas. Row annotations depict compound TCS discordance ratios for the reversal of AC-, MES-, NPC-, and OPC-like disease signatures calculated against non-neoplastic cells in the dataset.